# Ontogeny of circulating lipid metabolism in pregnancy and early childhood – a longitudinal population study

Satvika Burugupalli[1], Adam Alexander T Smith[1], Gavriel Oshlensky[1], Kevin Huynh[1], Corey Giles[1], Tingting Wang[1], Alexandra George[1], Sudip Paul[1], Anh Nguyen[1], Thy Duong[1], Natalie Mellett[1], Michelle Cinel[1], Sartaj Ahmad Mir[2,3], Li Chen[2,4], Markus R Wenk[2,3], Neerja Karnani[3,4], Fiona Collier[5,6], Richard Saffery[7,8], Peter Vuillermin[5,6,7], Anne-Louise Ponsonby[7,9], David Burgner[7,8]*, Peter Meikle[1]*, Barwon Infant Study Investigator team

[1]Metabolomics Laboratory, Baker Heart and Diabetes Institute, Melbourne, Australia; [2]Singapore Lipidomics Incubator, Life Sciences Institute, National University of Singapore, Singapore, Singapore; [3]Department of Biochemistry, Yong Loo Lin School of Medicine, National University of Singapore, Singapore, Singapore; [4]Singapore Institute for Clinical Sciences, A*STAR, Singapore, Singapore; [5]School of Medicine, Deakin University, Melbourne, Australia; [6]Child Health Research Unit, Barwon Health, Geelong, Australia; [7]Murdoch Children's Research Institute, Royal Children's Hospital, Melbourne, Australia; [8]Dept of Paediatrics, University of Melbourne, Parkville, Australia; [9]The Florey Institute of Neuroscience and Mental Health, Melbourne, Australia

*For correspondence:
david.burgner@mcri.edu.au (DB);
peter.meikle@baker.edu.au (PM)

Group author details:
Barwon Infant Study Investigator team See page 18

Competing interest: The authors declare that no competing interests exist.

## Abstract

**Background:** There is mounting evidence that in utero and early life exposures may predispose an individual to metabolic disorders in later life; and dysregulation of lipid metabolism is critical in such outcomes. However, there is limited knowledge about lipid metabolism and factors causing lipid dysregulation in early life that could result in adverse health outcomes in later life. We studied the effect of antenatal factors such as gestational age, birth weight, and mode of birth on lipid metabolism at birth; changes in the circulating lipidome in the first 4 years of life and the effect of breastfeeding in the first year of life. From this study, we aim to generate a framework for deeper understanding into factors effecting lipid metabolism in early life, to provide early interventions for those at risk of developing metabolic disorders including cardiovascular diseases.

**Methods:** We performed comprehensive lipid profiling of 1074 mother-child dyads in the Barwon Infant Study (BIS), a population-based pre-birth cohort and measured 776 distinct lipid features across 39 lipid classes using ultra-high-performance liquid chromatography-tandem mass spectrometry (UHPLC-MS/MS). We measured lipids in 1032 maternal serum samples at 28 weeks' gestation, 893 cord serum samples at birth, 793, 735, and 511 plasma samples at 6, 12 months, and 4 years, respectively. Cord serum was enriched with long chain poly-unsaturated fatty acids (LC-PUFAs), and corresponding cholesteryl esters relative to the maternal serum. We performed regression analyses to investigate the associations of cord serum lipid species with antenatal factors: gestational age, birth weight, mode of birth and duration of labour.

**Results:** The lipidome differed between mother and newborn and changed markedly with increasing child's age. Alkenylphosphatidylethanolamine species containing LC-PUFAs increased with child's age, whereas the corresponding lysophospholipids and triglycerides decreased. Majority of the cord serum lipids were strongly associated with gestational age and birth weight, with most lipids

showing opposing associations. Each mode of birth showed an independent association with cord serum lipids. Breastfeeding had a significant impact on the plasma lipidome in the first year of life, with up to 17-fold increases in a few species of alkyldiaclylglycerols at 6 months of age.

**Conclusions:** This study sheds light on lipid metabolism in infancy and early childhood and provide a framework to define the relationship between lipid metabolism and health outcomes in early childhood.

**Funding:** This work was supported by the A*STAR-NHMRC joint call funding (1711624031).

## Editor's evaluation

The paper is well written and the experimental approach is sound. It contains a large amount of data and addresses an important and an understudied area of science. The authors were very responsive to the reviewers' comments and the reviewers unanimously decided that the paper is significantly improved and is ready for publication. The paper will likely have a high impact on the field, as it identifies changes in plasma lipids at the early stages after birth and its link with factors related to pregnancy.

## Introduction

The developmental origins of health and disease paradigm suggests that prenatal, perinatal, and postnatal heritable and environmental influences result in long-term developmental, physiological, and metabolic changes in major tissues and organs (*Barker et al., 1993*; *Barker, 1999*). These metabolic perturbations can contribute to later life disease risk, including cardiovascular diseases and related cardiometabolic conditions (*Barouki et al., 2012*; *Barker, 2007*). Alterations to lipid metabolism are a key driver of metabolic disorders (*Castro Cabezas et al., 2018*; *Meikle and Summers, 2017*). The lipid profile in blood (the lipidome) provides an integrative measure of genetic and environmental exposures on circulating lipid metabolism and quantification of the lipidome offers a rapid and effective approach to identify biomarkers potentially predictive of several diseases (*Yang et al., 2016*). Large population-based studies from our group and others have established specific lipids to be associated with cardiometabolic disorders including diabetes and cardiovascular diseases (*Huynh et al., 2019*; *Weir et al., 2013*; *Huynh et al., 2020*; *Lamichhane et al., 2018*). In contrast to adult data, little is known about lipid metabolism in early life. Understanding the key determinants of early life lipid metabolism will inform the development of risk-stratification and early interventions.

The Barwon Infant Study (BIS) is a longitudinal Australian birth cohort with antenatal recruitment and repeated in-depth data and sample collection that is designed to facilitate a detailed mechanistic investigation of disease development within an epidemiological framework (*Vuillermin et al., 2015*). In this study, we applied a comprehensive lipidomic approach to understand the lipid metabolism during pregnancy and throughout the first 4 years of life. We utilised targeted ultra-high-performance liquid chromatography-tandem mass spectrometry (UHPLC-MS/MS) to measure 776 distinct individual lipid species across 39 distinct lipid classes (ceramide and deoxyceramide have been grouped together). This aimed to increase understanding of the lipid metabolism in newborns and infants in the first year of life, specially focusing on the association of cord serum lipids with antenatal factors: gestational age, birth weight, mode of birth, and duration of labour. We also aimed to investigate the relationship between breastfeeding and plasma lipids at 6 and 12 months of age. We provide a baseline characterisation of circulating lipids in the mother and newborn and changes in plasma lipids from birth to 4 years. In summary, the results from this study provide a framework for future studies to understand lipid metabolism in early life and provide early interventions for lipid dysregulation.

## Materials and methods
### Research design and the cohort

The BIS is a pre-birth cohort (n = 1074 mother-infant pairs, 72% Anglo-European) from the Barwon region of south-eastern Australia (*Vuillermin et al., 2015*). Women were recruited prior to 28 weeks of pregnancy. Infants born prior to 32 weeks, who developed a serious illness in the first week of life,

or who had significant congenital or genetic abnormalities, were excluded. Participants were reviewed at birth and at 1, 6, 9, 12, 24, and 48 months. Maternal serum samples were collected at 28 weeks' gestation. Child serum or plasma samples were collected at birth, 6 months, 12 months, and 4 years. Data on maternal age, birth order, prenatal weight, and antenatal comorbidities were collected from questionnaires and hospital records, and by standardised clinical examination. Cord blood was collected at birth in a serum clotting tube, samples were centrifuged within 2 hr of collection, and the serum was separated and stored at –80°C. Maternal pre-pregnancy body-mass index (BMI) was calculated from self-reported pre-pregnancy weight and directly measured maternal height at the first study visit (28–32 weeks' gestation). Birth anthropometric measures (birth weight, length, and head circumference) were collected within the first 2 days of life. Anthropometric measurements were also obtained at birth, 6 months, 12 months, and 4 years. BMI at each time point was calculated by weight (kilograms)/height (metres)$^2$. Ethics approval for this study was granted by the Barwon Health Human Research and Ethics Committee (HREC 10/24).

## Lipidomic profiling

### Lipid extraction

Lipid extraction was performed as described previously in *Alshehry and Meikle, 2016*. In brief, 10 µL of plasma was mixed with 100 µL of butanol:methanol (1:1) with 10 mM ammonium formate containing a mixture of internal standards. Samples were vortexed, sonicated for an hour, and then centrifuged (14,000× *g*, 10 min, and 20°C) before transferring the supernatant into sample vials with glass inserts for analysis.

### Liquid chromatography-tandem mass spectrometry

Lipidomics was performed as described previously in *Weir et al., 2013* and *Huynh et al., 2019*, with adaptations for a dual column setup. Analysis of plasma extracts was performed on an Agilent 6490 QQQ mass spectrometer with an Agilent 1290 series UHPLC system and two ZORBAX eclipse plus C18 column (2.1 × 100 mm, 1.8 mm, Agilent) with the thermostat set at 45°C. Mass spectrometry analysis was performed in both positive and negative ion mode with dynamic scheduled multiple reaction monitoring (MRM). A detailed list of MRMs and internal standards in provided in *Supplementary file 1A*.

The running solvent consisted of solvent A: 50% $H_2O$/30% acetonitrile/20% isopropanol (v/v/v) containing 10 mM ammonium formate and solvent B: 1% $H_2O$/9% acetonitrile/90% isopropanol (v/v/v) containing 10 mM ammonium formate. To avoid peak tailing for acidic phospholipids, we passivised the instrument prior to each batch by running 0.5% phosphoric acid in 90% acetonitrile for 2 hr and subsequently flushing the system with 85% $H_2O$/15% acetonitrile prior to sample run.

We utilised a stepped linear gradient with a 12.9 min cycle time per sample and a 1 µL sample injection. To increase throughput, we used a dual column setup to equilibrate the second column while the first is running a sample. The sample analytical gradient was as follows: starting with a flow rate of 0.4 mL/min at 15% B and increasing to 50% B over 2.5 min, then to 57% over 0.1 min, to 70% over 6.4 min, to 93% over 0.1 min, to 96% over 1.9 min, and finally to 100% over 0.1 min. The solvent was then held at 100% B for 0.9 min (total 12.0 min). Equilibration was started as follows: solvent was decreased from 100% B to 15% B over 0.2 min and held for an additional 0.8 min for a total run time of 12.9 min per sample. The next sample is injected, and the columns are switched.

The following mass spectrometer conditions were used: gas temperature, 150°C, gas flow rate 17 L/min, nebuliser 20 psi, sheath gas temperature 200°C, capillary voltage 3500 V, and sheath gas flow 10 L/min. Isolation widths for Q1 and Q3 were set to 'unit' resolution (0.7 amu).

Plasma quality control (PQC) samples consisting of a pooled set of plasma samples taken from six healthy individuals and extracted alongside the study samples were incorporated into the analysis at 1 PQC per 20 plasma samples. Technical QC samples (TQC) consisted of PQC extracts which were pooled, then split into individual vials to provide a measure of technical variation from the mass spectrometer only. These were included at a ratio of 1 TQC per 20 plasma samples. TQCs were monitored for changes in peak area, width, and retention time to determine the performance of the UHPLC-MS/MS analysis and were subsequently used to account for differential responses across the analytical batches. To align the results to any future datasets, the NIST 1950 SRM sample (Sigma) was included as a reference plasma sample, at a rate of 1 per 40 samples.

Relative quantification of lipid species was determined by comparison to the relevant internal standard. Lipid class total concentrations were calculated as the sum of individual lipid species concentrations, except in the case of triacylglycerol (TG) and alkyldiacylglycerol (TG(O)) species, where we measured both neutral loss (NL) and single ion monitoring (SIM) peaks, and subsequently used the abundant but less structurally resolved (SIM) species concentrations for summation purposes when examining lipid class totals. Any lipid differences based on the matrix must be kept in mind as serum samples were used during gestation, and at birth vs. plasma samples at other time points. Several studies have reported matrix-associated differences in species of lysophospholipids, diacylglycerols, free long chain fatty acids, and oxidised fatty acids (*Ishikawa et al., 2014*; *Sangkuhl et al., 2011*). However, these differences are unlikely to reflect biological processes in the body (*Lagarde et al., 2010*). These differences could be reflected in change in lipid levels observed at birth and 6 months, but not affect the association with outcomes.

## Data processing

Lipidomic analysis was performed across 11 batches of ~500 samples with quality control samples analysed every 20 samples. The chromatographic peaks were integrated using the Mass Hunter (B.07.00, Agilent Technologies) software and assigned to specific lipid species based on MRM (precursor/product) ion pairs and retention time. Upon completion of quantification, lipidome data for 777 lipid features in 4007 samples was available across 11 batches. Zero values (values beneath the mass spectrometer's detection threshold) were replaced by 1/10th the minimum value for the concerned lipids in the corresponding sample types. We then aligned the lipid concentrations across different batches by median centring the PQC samples, and additionally scaled all batch-wise standard deviations for biological samples (on the log scale) to the mean standard deviation. We visually assessed lipids for large-scale technical variations (such as groups of outlier measurements, effects of drift or lipid hydrolysis over batch run duration), and re-quantified affected lipids where possible, and excluding them or selectively setting some values to missing where not. Missing values were then imputed using a k-nearest neighbour approach in the samples space. We used two different approaches for outlier sample detection. On one hand, lipid concentrations were log-transformed, and Z-scored, and absolute Z-scores were summed for each sample as a measure of their extremeness. Samples showing sums above the 95th percentile were flagged as potential outliers. On the other hand, the Z-scores were used in a principal components analysis (PCA), and the seven first principal components were retained as explaining the most variance. In the space defined by these components, each sample's distance to the origin was calculated as a measure of the sample's extremeness. Samples with a distance greater than the 95th percentile were flagged as potential outliers. Samples flagged by both approaches were declared outliers are removed from further analysis. Further inspection revealed that 16 samples were 'missed injections' on the mass spectrometer, 27 cord serum samples were contaminated by maternal blood, these which were excluded, leaving 3964 samples in total. Ultimately, we retained 776 lipid measures across all batches. SIM-based lipid measures were not included in lipid species-level analyses, leaving 733 species.

## Statistical analysis

All statistical analyses were carried out in R (3.5.0 or 4.0.3). Plasma and cord serum lipid concentrations were log10-transformed prior to statistical analysis. PCA was performed using package FactoMineR using default settings.

Associations between maternal/infant/children characteristics and lipid species were determined using multiple linear regression, adjusting for appropriate covariates in each analysis (*Lê et al., 2008*). All the analyses at birth using cord serum were adjusted for birth weight, gestational age, mode of birth, duration of labour, sex, maternal pre-pregnancy BMI, maternal gestational diabetes mellitus (GDM), maternal education, maternal age, birth order, and lipidomics run batch.

Longitudinal analyses (associations of lipid species with breastfeeding, sex, and weight across infancy and early childhood) were performed using linear regression models of breastfeeding/sex/weight against the log-transformed abundance of lipid species at each time point, adjusted for sex and age. Breastfeeding status (y/n) at 6 and 12 months were taken from questionnaires at these time points. We observed that gestational age influences lipids at 6 months and not at 12 months and hence adjusted for gestational age at the 6-month time point. The effective sample size varied slightly

with each time point and between models due to missing samples or missing values for some covariates. Beta-coefficients and 95% confidence intervals were then converted to percentage difference (percentage change = $(10^\beta\text{-coefficient} - 1) \times 100$) to facilitate interpretation of results. All p-values were corrected for multiple comparisons, per model term, using the method of *Benjamini and Hochberg, 1995*. Differences between the maternal antenatal plasma lipidome and the cord serum lipidome were tested using paired Wilcoxon ranked sum tests.

## Data representation

The results from the association analyses are presented as forest plots in Figures 2–4 and 6, where each lipid species (or class) is represented by a marker showing either a positive association (increases with an increase in the outcome) or a negative association (decreases with an increase in the outcome). The colour of the markers relate to the significance of the association as indicated in the figure legends. The lipid species are grouped into lipid class and subclass and ordered into categories (sphingolipids, phospholipids, and other lipids) each separated by a grey dotted line.

# Results

The baseline characteristics of the complete cohort are shown in *Supplementary file 1B*. In this study, we utilised serum samples of the mothers at 28 weeks' gestation, cord serum samples at birth, and plasma samples at 6, 12 months, and 4 years.

## Serum/plasma lipidomics

We measured the plasma lipid species in a total of 3964 samples: 1032 in mothers, 893 at birth, 793 at 6 months, 735 at 12 months, 511 at 4 years (*Figure 1a*). Of the measured lipid species and classes, we retained 733 lipid species across 37 classes for further analyses. The median coefficient of variation (CV) for the lipid measures based on the PQC samples was 10.1%, with 92.6% of lipid measures showing a CV <20%.

## The serum/plasma lipidome differs between mothers, newborns, and infants

Overall lipid levels increased with age: maternal serum had higher total lipid levels than infant plasma, which in turn had higher levels than cord serum (*Figure 1b*). PCA of the lipidomic data from all participants at all the time points revealed a clear separation of maternal, newborn, and infant samples across the first and second principal components (*Figure 1c*). A PCA of the infant samples showed further separation of the 6-, 12-, and 4-year time points (*Figure 1d and e*).

## Gestational age and birth weight influence newborn cord serum lipids

Of 733, 461 (63%) cord lipid species were associated with gestational age and 299 (of 733, 41%) lipid species were associated with birth weight (*Figure 2a and b*). *Figure 2—figure supplement 1* explains lipid metabolism pathways. Gestational age was modestly correlated with birth weight (r = 0.45). However, the lipid profile associated with birth weight was discordant with the lipid profile associated with gestational age as indicated by the plot of the beta-coefficients for gestational age against the beta-coefficients for birth weight (y = –10.535x + 0.0101, $r^2$ = 0.714, *Figure 2—figure supplement 2*). Of 733, 591 (80%) lipid species had an opposing association with gestational and birth weight. Species of di- and triacylglycerol, alkyldiacylglycerol, acylcarnitine, and free fatty acids (FFAs) increased with gestational age but decreased with birth weight. Phospholipid species containing either an odd-numbered straight, or a methyl branched fatty acid (15-methylhexadecanoic acid [15-MHDA]) were positively associated with gestational age (*Figure 2—figure supplement 3a*) but negatively associated with birth weight (*Figure 2—figure supplement 3b*). While lysophospholipid species as a class were negatively associated with gestational age, we observed elevated levels of odd- and branched-chain fatty acid containing lysophosphatidylcholine species: LPC (19:0) [sn1] and [sn2], LPC (15-MHDA) [sn1] and [sn2] with gestational age that had an opposite association with birth weight. Cholesteryl esters (CE) also showed a similar trend as seen by elevated levels of CE(15:0) and CE(17:0). Alkyl- and alkenylphosphatidylethanolamine species increased with both gestational age and birth weight. Species of lysophospholipids, alkyl- and alkenylphosphatidylcholine, dihydroceramide, hexosylceramide,

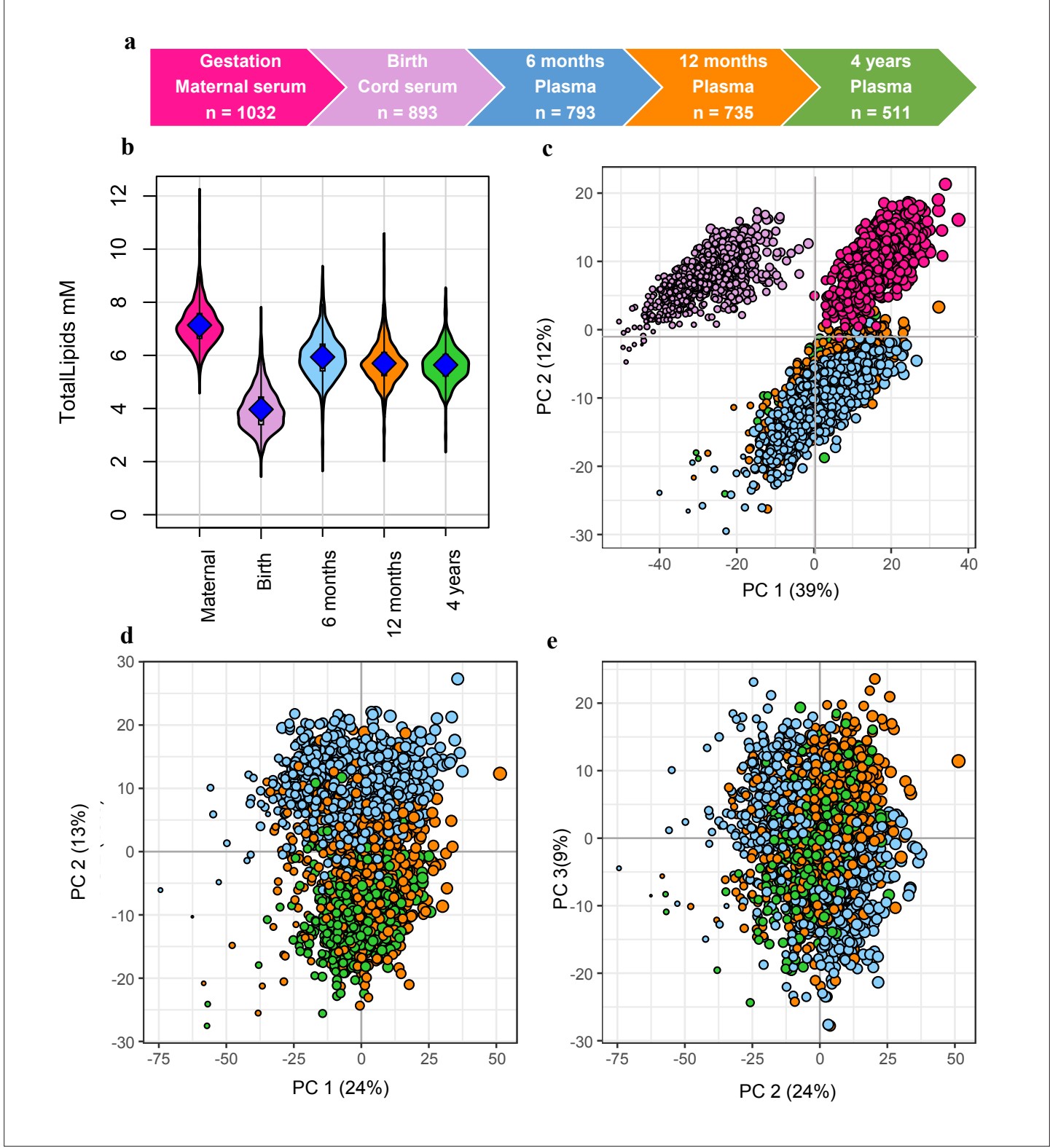

**Figure 1.** Snapshot of the Barwon Infant Study (BIS) lipidomics. (**a**) The BIS timeline and final lipidomic sample numbers at each time point. (**b**) Total lipid concentration in maternal and infant sample groups. (**c**) Principal component analysis (PCA) of the lipidomic measures at all the time points, marker size is proportional to the median log total lipid level (%). (**d**) PCA (PC1 vs. PC2) of the lipidomic measures at 6 months, 12 months, and 4 years, marker size as previously stated. (**e**) PCA (PC2 vs. PC3) of the lipidomic measures at 6 months, 12 months, and 4 years, marker size as previously stated. For all panels: Maternal serum, M28, pink colour; newborn cord serum, birth, purple colour; 6-month plasma, blue colour; infant 12-month plasma, orange colour; 4-year plasma, green colour.

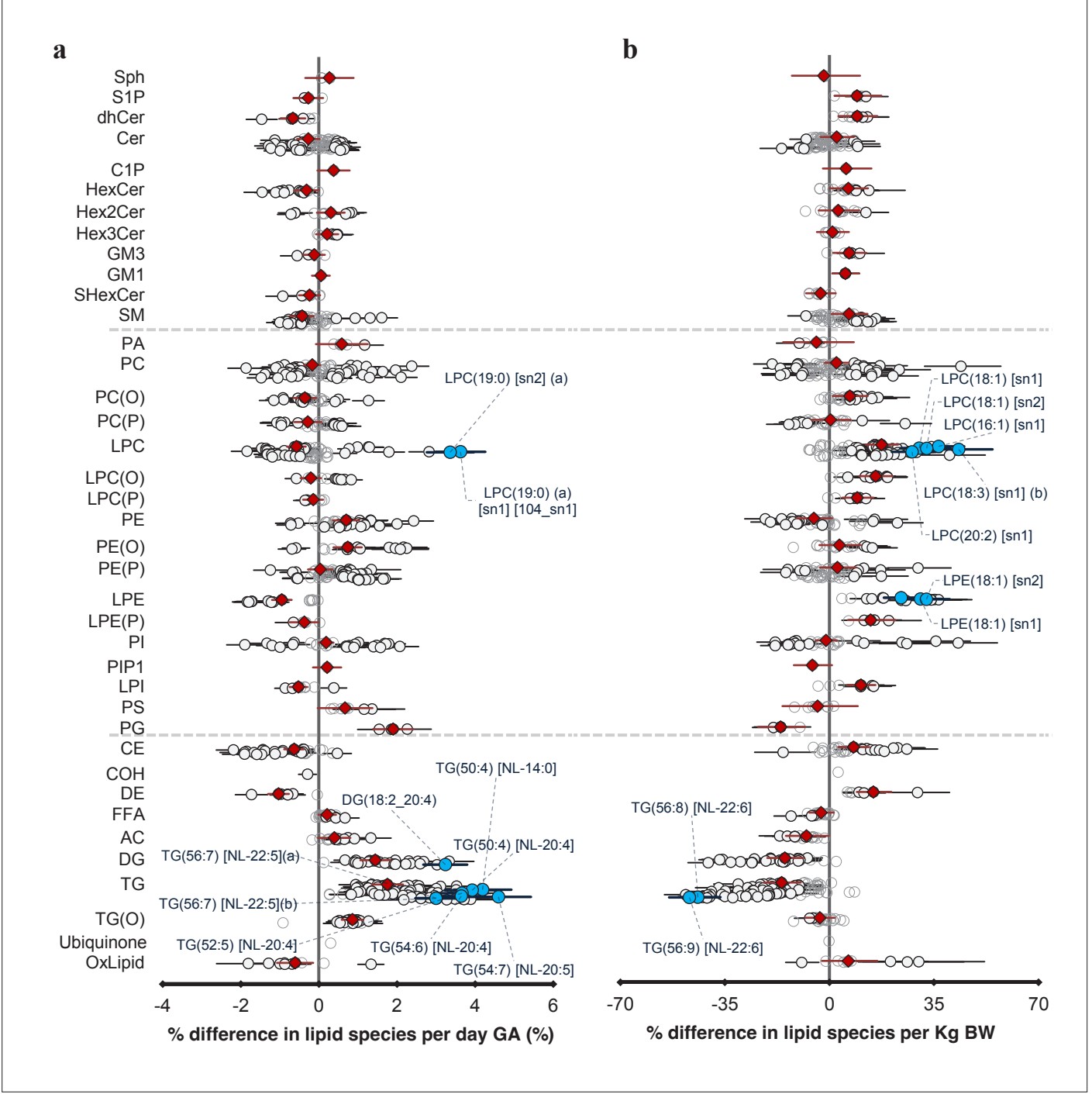

**Figure 2.** The newborn cord serum lipidome is influenced by gestational age (GA) and birth weight (BW). ( a) Estimated percentage difference of the lipid species per day increase in gestational age, determined using linear regression, adjusted for weight, sex, mode of birth, duration of labour, maternal pre-pregnancy BMI, gestational diabetes mellitus (GDM), maternal education, birth order, and lipidomics run batch. (b), Estimated percentage difference of the individual lipid species per kilogram increase in birth weight, determined using the same model as (**a**).

The online version of this article includes the following figure supplement(s) for figure 2:

**Figure supplement 1.** Schematic representation of the lipid metabolism and related pathways.

**Figure supplement 2.** Relationship between the associations of lipid species with gestational age and birth weight.

**Figure supplement 3.** Association of cord serum phospholipids with gestational age and birth weight.

cholesteryl ester, and dehydrocholesteryl ester decreased with gestational age and increased with birth weight. Complete associations between gestational age, birth weight, and gender with lipid species and lipid classes at birth are detailed in *Supplementary file 1C and D*.

Each circle on the plot represents a lipid species, grey open circles show non-significant lipid species (p > 0.05), white closed circles show significant lipid species (p < 0.05), the top 10 lipid species (10 lowest p-values) are shown in blue circles. Each red diamond represents a lipid class. All p-values were corrected for multiple comparisons. The horizontal bars (error bars) are shown for the significant lipid species (white closed circles), the top 10 lipid species (blue circles), corrected p-value < 2.88 × $10^{-24}$ and 1.81 × $10^{-15}$ respectively and the lipid classes (red diamonds). The error bars represent 95% confidence intervals. Grey dotted lines separate sphingolipids, phospholipids, and other lipids.

## Mode of birth and duration of labour are associated with specific signatures in the cord serum lipidome

In this study, we compared four different birth modes: assisted vaginal birth (assisted VB, n = 214) (including both forceps- and vacuum-assisted birth), scheduled caesarean (CS, n = 178) and unscheduled caesarean (unscheduled CS, n = 155), with unassisted vaginal birth (VB, n = 525) as reference. Unscheduled CS accounted for all the emergency CS including breech births, while individual data on the reason for the emergency was not recorded. Of 733, 214 lipid species (29%) showed evidence of association with at least one mode of birth (*Figure 3*). Lipid species of acylcarnitine, di- and triacylglycerols were elevated in both assisted VB and unscheduled CS relative to VB, while lipid species of FFAs, lysophosphatidylcholine decreased in both CS and unscheduled CS birth modes compared to VB (*Figure 3a, b, and c*, and *Supplementary file 1E and F*). Unscheduled CS birth had an intermediate profile where 74% of the lipid species showed a similar association to that observed with assisted VB ($r^2$ = 0.543, *Figure 3—figure supplement 1*), while 57% of the lipid species showed the same trend as observed with the scheduled CS birth mode.

Of 733, 36 lipid species (5%) showed evidence of association with duration of labour independent of the mode of birth. Similar to assisted VB and unscheduled CS, we observed elevated levels of acyl-carnitine, di- and triacylglycerols with longer duration of labour.

We then investigated the association of maternal lipids at 28 weeks' gestation with mode of birth and duration of labour. However, none of the maternal lipids were significantly associated with any of the delivery modes or the duration of labour (*Supplementary file 1G and H*).

## Long chain polyunsaturated fatty acids are enriched in newborn cord serum

We performed paired Wilcoxon tests to investigate the relationship between maternal and newborn lipidome. While 80% of the lipid species were significantly higher in the maternal serum, the lipid species of sphingosine and sphingosine-1-phosphate, phosphatidylserine, acylcarnitine, dehydro-cholesteryl ester, glycosphingolipids, and FFAs were higher in the newborn lipidome (*Figure 4a*). Omega-3 and omega-6 long chain polyunsaturated fatty acids (LC-PUFAs): arachidonic acid (20:4, n-6), eicosapentanoic acid (20:5, n-3), adrenic acid (22:4, n-6), docosapentaenoic acid (DPA, 22:5, n-6), docosahexaenoic acid (DHA, 22:6, n-3) were enriched in the cord serum (*Figure 5b and c*). While lipid classes such as cholesteryl ester, lysophosphatidylcholine, and triglycerides were higher in the maternal plasma individual lipid species containing LC-PUFAs were higher in cord serum (*Figure 4b and c*, *Supplementary file 1I and J*).

## The circulating lipidome changes over the first 4 years

We performed paired t-tests to study the changes in lipid concentration at different time points (*Figure 5*). Of 733, 549 (75%) of the lipid species were higher at 4 years than at birth, with LC-PUFA containing alkenylphosphatidylethanolamine species having the greatest increase. Of 733, 123 lipid species (17 %) consistently increased at all the time points from birth to 4 years, whereas 68 lipid species decreased across all the time points. There was a consistent decrease in LC-PUFA containing lysophospholipids and an increase in odd- and branched-chain fatty acids, and phospholipids containing these fatty acids, across all the time points.

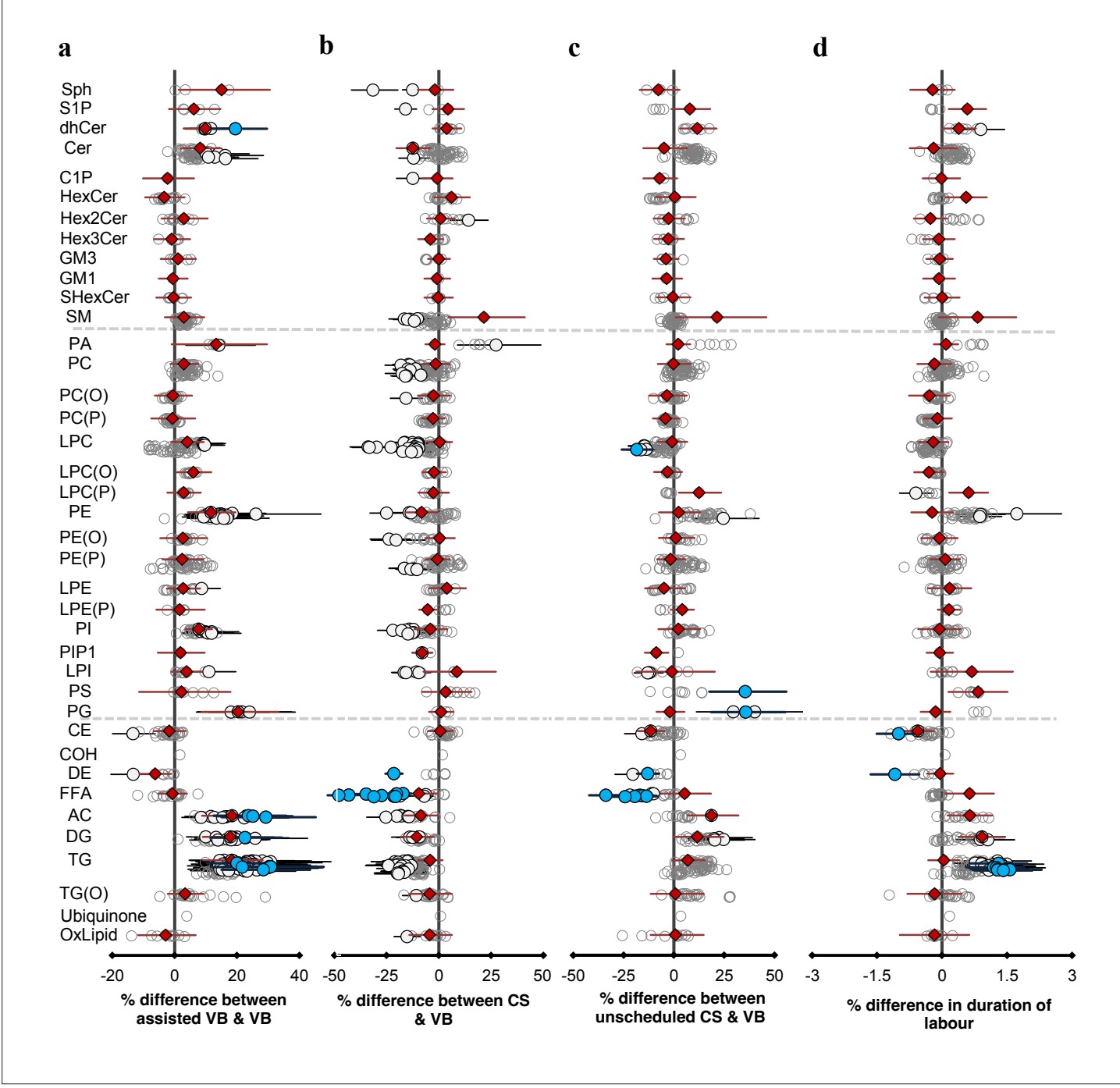

**Figure 3.** Mode of birth influences the newborn cord serum lipidome. Linear regression of the lipidome on delivery mode, each mode: assisted vaginal birth (assisted VB), scheduled caesarean (CS), unscheduled caesarean (unscheduled CS) being compared to unassisted vaginal birth (VB), adjusting for birth weight, gestational age, sex, duration of labour, maternal pre-pregnancy BMI, gestational diabetes mellitus (GDM), maternal education, birth order, and lipidomics run batch. Results are shown as % difference in lipids between each mode and VB. (**a**) Lipid species associated with assisted vaginal delivery. (**b**) Lipid species associated with scheduled caesarean delivery. (**c**) Lipid species associated with unscheduled caesarean delivery. (**d**) Lipid species associated with duration of labour, independent of the mode of delivery. Each circle on the plot represents a lipid species, grey open circles show non-significant lipid species (p > 0.05), white closed circles show significant lipid species (p < 0.05), the top 10 lipid species (10 lowest p-values) are shown in blue circles. Each red diamond represents a lipid class. All p-values were corrected for multiple comparisons. The horizontal bars (error bars) are shown for the significant lipid species (white closed circles), the top 10 lipid species (blue circles), Corrected p-value < 1.64 × 10−03, 2.16 × 10−12, 2.27 × 10−03, 1.8 × 10−02, respectively, and the lipid classes (red diamonds). The error bars represent 95% confidence intervals. Grey dotted lines separate sphingolipids, phospholipids, and other lipids.

*Figure 3 continued on next page*

*Figure 3 continued*

The online version of this article includes the following figure supplement(s) for figure 3:

**Figure supplement 1.** Comparison of beta coefficients of lipids associated with assisted vaginal birth vs unscheduled and scheduled caesarean births.

### Birth to 6 months

Majority of lipid species (558 of 733, 76%) were higher at 6 months compared to at birth. We observed the greatest increase in species of alkenylphosphatidylethanolamine and alkyldiacylglycerol and the largest decrease in 20:4 and 20:6 containing lysophosphatidylcholine, lysophosphatidylethanolamine, and cholesteryl esters (*Supplementary file 1K and L*).

### Six to 12 months

Of 733, 363 (50%) lipid species increased in concentration from 6 to 12 months. Odd- and branched-chain phospholipids showed the greatest increase while species of alkyldiacylglycerol and cholesteryl esters showed the greatest decrease (*Supplementary file 1K and L*).

Of 733, 272 (37%) lipid species consistently increased from birth to 6 months and from 6 to 12 months, while 90 (of 733, 12%) lipid species showed a consistent decrease. Of 733, 365 (50%) lipid species had an opposing trend at these time points. Lipid species of dihydroceramide, phospholipids, glycophospholipids, lysophospholipids, FFA, alkyl- and alkenylphospholipids, di- and triacylglycerols, acylcarnitine, sphingomyelin, mono- and trihexosyl ceramides, ceramide-1-phosphate increased from birth to 6 months but decreased from 6 to 12 months. Lipid species of sphingosine decreased from birth to 6 months but increased from 6 to 12 months.

### Twelve months to 4 years

Of 733, 284 (37%) of the lipid species were higher at 4 years than at 12 months. We saw the highest increase in CE 20:5 and largest decrease in 22:6 containing di- and triacylglycerols (*Supplementary file 1K and L*).

Some lipid species showed variable changes as the child aged. Lipid species of dehydrocholesterol, alkenyllysophosphatidylcholine that decreased from birth to 6 months, and from 6 to 12 months, increased at 4 years. Lipid species of ceramide, phosphatidylethanolamine, and phosphatidylglycerol that increased from birth to 6 months and 6–12 months, decreased at 4 years. Lipid species of sphingosine decreased from birth to 6 months, increased at 12 months, and decreased again at 4 years (*Figure 5—figure supplement 1*).

Overall omega-3 and omega-6 LC-PUFAs: 20:3, 20:4, 22:4, 22:5, and 22:6 that were higher at birth, decreased by the time the infant reached 6 months, and continued decreasing further until 4 years. Most of the omega-3 PUFA cholesteryl esters: CE (22:5) (n-3) and 22:6 decreased in the first year but increased at 4 years except for CE (20:5) that decreased at 6 months but increased at 12 months and 4 years. Omega-6 PUFA cholesteryl esters: CE (20:4), CE (22:4), and CE (22:5) (n-6) decreased at 6 months but increased at 12 months and 4 years (*Supplementary file 1M*).

Additionally, we performed time-series clustering to capture the changes of all the lipid species across all the time points. We utilised dynamic time warping distance as a dissimilarity measurement (*Aghabozorgi et al., 2015*). We used the partitional clustering method with partitions around medoids to cluster the lipid species. We identified 10 different lipid clusters, each of them showing unique lipid trajectories over the first 4 years of life (*Figure 5—figure supplement 2*, *Supplementary file 1K*).

Cluster 1, consisted of 63 lipid species, comprising of ceramide species and LC-PUFA containing phospholipid species, all of which increased from birth to 6 months, and then stabilised by the time the child reached 12 months. Cluster 2 consisted of 149 lipid species of which phospholipids such as the odd chain phosphatidylcholine PC(31:0) and branched-chain and LC-PUFA containing phosphatidylethanolamine species: PE(15-MHDA_20:4) and PE(15-MHDA_22:6) decreased at 6 months, increased at 12 months, and decreased at 4 years. These PE species were also positively associated with birth weight in cord serum. Lysophospholipids containing 18:1 and 18:2 fatty acids, increased at 6 months, decreased at 12 months and increased as the child reached 4 years. In adults, these lysophospholipids are generally associated with higher BMI. LC-PUFA containing phospholipids, especially alkenylphosphatidylethanolamine and alkylphosphatidylcholine, increased at 6 months, but decreased at 12 months and 4 years. Similarly, alkyldiacylglycerols increased at 6 months and decreased thereafter.

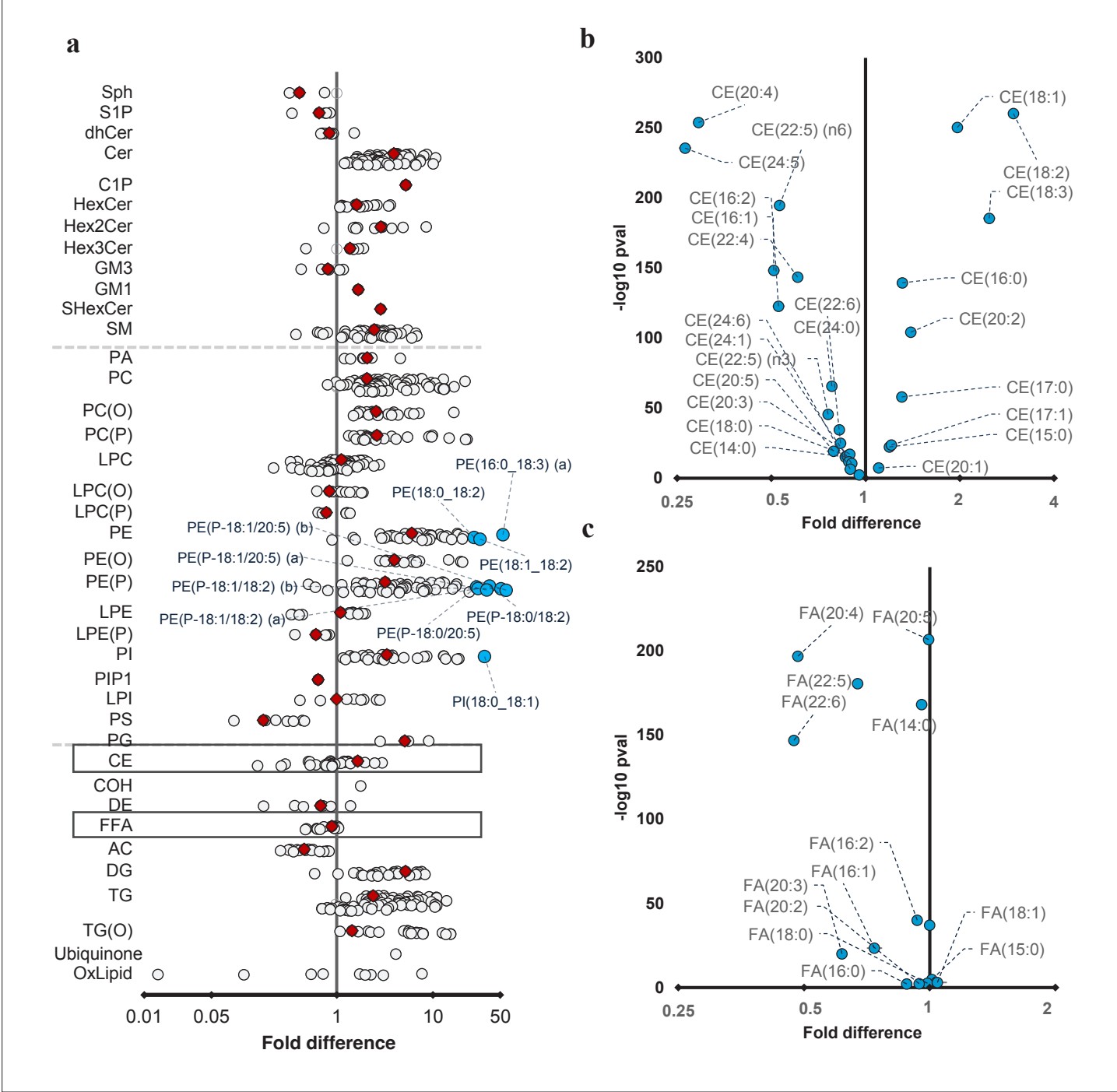

**Figure 4.** Differences in maternal and cord serum lipid species. Paired Wilcoxon tests were performed, and results are shown as fold differences of mothers relative to infants. (**a**) Fold differences in lipid concentrations between maternal and cord serum. Cholesteryl esters and free fatty acid (FFA) classes are highlighted in boxes. (**b**) Selective enrichment of cholesteryl esters containing long chain polyunsaturated fatty acids in cord serum. (**c**) Selective enrichment of long chain polyunsaturated fatty acids in cord serum. Each circle on the plot represents a lipid species, grey open circles show non-significant lipid species ($p > 0.05$), white closed circles show significant lipid species ($p < 0.05$), the top 10 lipid species (10 lowest p-values) are shown in blue circles. Each red diamond represents a lipid class. All p-values were corrected for multiple comparisons. The horizontal bars (error bars) are shown for the significant lipid species (white closed circles), the top 10 lipid species (blue circles), corrected p-value $< 5.06 \times 10^{-261}$, and the lipid classes (red diamonds). The error bars represent 95% confidence intervals. Grey dotted lines separate sphingolipids, phospholipids, and other lipids.

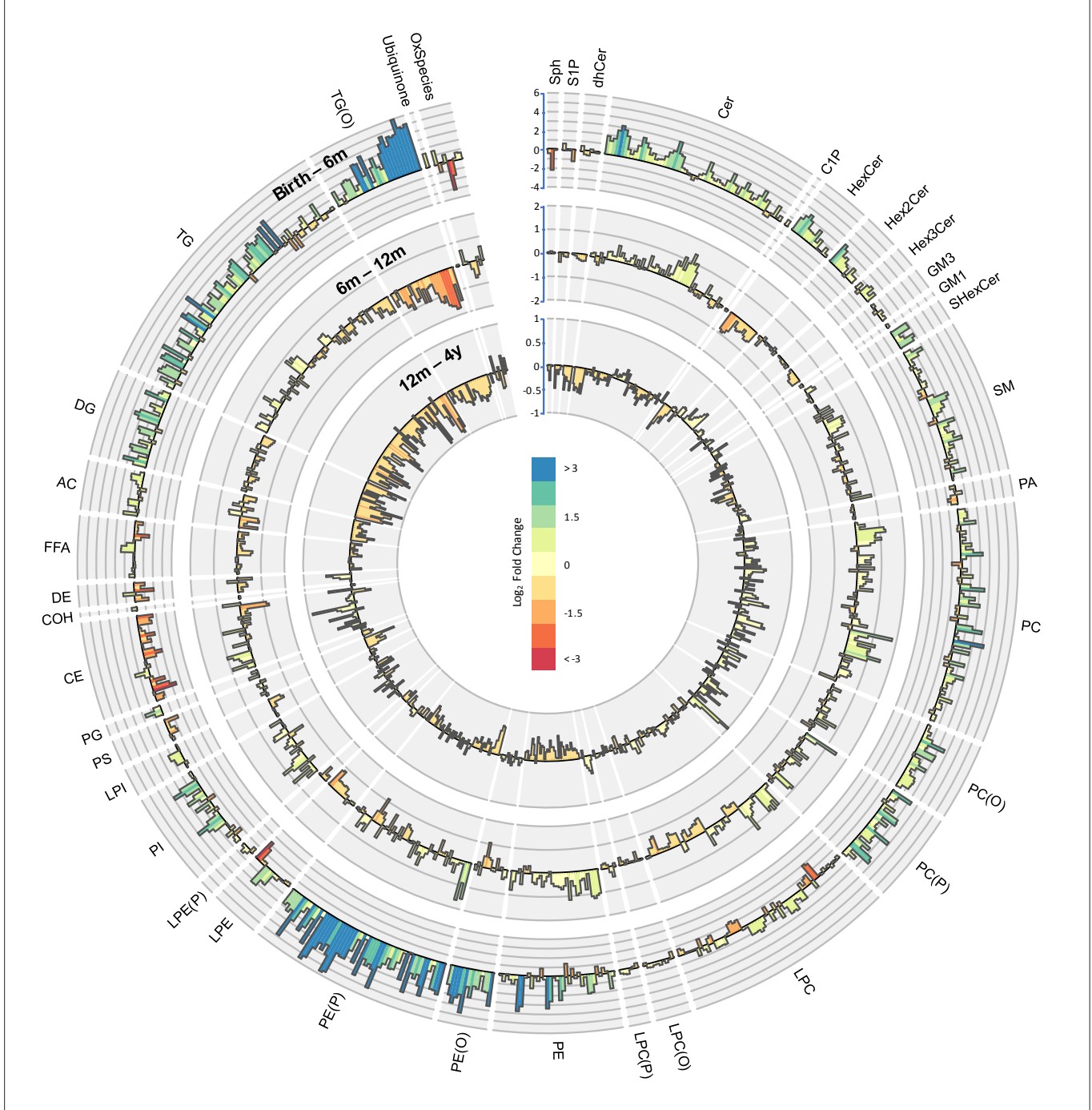

**Figure 5.** Lipidomic profile changes in the first 4 years. Paired t-tests were performed in infants with repeat measures (birth to 6 months n = 646, 6–12 months n = 628, 12 months to 4 years n = 418), and results are shown as log2 fold change (FC). The outermost circle represents the changes in lipid concentration from birth to 6 months (axis range: –4 to 6), middle circle represents changes in lipid concentration 6–12 months (axis range: –2 to 2), and the inner most circle represents the changes in lipid concentration from 12 months to 4 years (axis range: –1 to 1).

The online version of this article includes the following figure supplement(s) for figure 5:

**Figure supplement 1.** The serum/plasma concentration of lipid classes over time.

**Figure supplement 2.** Time series clustering on lipid species identified 10 different lipid clusters.

Cluster 3 comprised of 143 lipid species. Species of phospholipids in cluster 3 increased at all time points while ceramide and phospholipid species containing odd chain fatty acids increased until 12 months and decreased by the time the child reached 4 years. Cluster 4 consisted of 41 lipid species. The pattern was similar to cluster 2, with species of LC-PUFA containing phosphatidylethanolamine species increasing from birth to 6 months and decreasing at 12 months and 4 years. Cluster 5 consists of 60 lipid species. Most of the lipid species in cluster 5 increased until the child reached 6 months but decreased as the child reached 12 months and further. Cluster 6 consisted of 38 lipid species, mostly exhibiting mixed patterns. Cluster 7 consisted of 51 lipid species. As seen in cluster 2, the alkyldia-cylglcyerols increased at 6 months and decreased at 12 months and 4 years, but the alkenylphospha-tidylethanolaminespecies increased until 12 months and decreased at 4 years. Cluster 8 consisted of 33 lipid species, with species of lysophospholipids, cholesteryl esters, and fatty acids showing a consistent decrease across all time points. Cluster 9 consisted of 98 lipid species and showed a similar profile to cluster 1. Ceramide and sphingomyelin species containing 23:0 and 24:0 long chain fatty acids increased across all the time points. Additionally, alkyl- and alkenylphosphatidylethanolamine species containing LC-PUFAs (18:2, 20:3, and 20:4) increased across all the time points. Cluster 10 consisted of 57 lipid species. Fatty acids 20:0 and 22:0 containing hexosylceramides and lysophos-phatidylcholines followed a peculiar pattern of increasing until 6 months decreasing at 12 months, and later increasing as the child reaches 4 years. Acylcarnitine and fatty acid species, that were elevated in cord serum, compared to maternal serum, increased until 6 months and later decreased at 12 months and further at 4 years suggesting a change in energy metabolism over the first 4 years.

## Breastfeeding impacts lipid metabolism in the first year of life

We investigated the association of breastfeeding with plasma lipids at 6 and 12 months. Of the 776 lipid species, 664 lipid species (90%) were significantly associated with breastfeeding at 6 months, and 438 (of 773, 65%) were significantly associated at 12 months (*Supplementary file 1N and O*). Of particular note, species of alkyl- and alkenylphospholipids (plasmalogens) and alkyldiacylglycerides (TG-O) were markedly elevated in breastfed infants. At a class level these elevations were of the order of two- to fourfold, while individual species of alkyldiacylglycerols: TG(O-54:2) [NL-17:1], TG(O-54:2) [NL-18:1], TG(O-52:1) [NL-18:1] were elevated 17- to 19-fold at 6 months. However, at 12 months, this effect size was only two- to fourfold, given the introduction of a wide variety of foods along with breast milk (*Figure 6*). In addition to the dramatic increase in these ether lipid species, we also observed an increase in many species containing odd- and branched-chain fatty acids such as phos-phatidylcholine PC(33:0), sphingomyelin SM(d19:1/24:1), and others. These odd- and branched-chain fatty acids represent a novel, potentially bioactive, class of lipids enriched in breastfed infants.

## Sex-specific changes in the circulating lipidome over the first 4 years

We investigated the influence of sex on lipid metabolism in early childhood. We found 218 lipid species (of 733, 29%) to be significantly associated at birth, 214 (of 733, 29%) at 6 months, 62 (of 733, 8%) at 12 months, and 82 (of 733, 11%) at 4 years. We found significant differences in lipid species of ceramide, sphingomyelin, acylcarnitine, lysophosphatidylcholine, lysophosphatidylethanolamine, alkyllysophosphatidylcholine, and triacylglycerols between boys and girls (*Supplementary file 1P and Q*).

## Discussion

This is the first and largest study of the lipidome from birth and early childhood in a population-derived cohort. The lipidome was different between mother and child and changed markedly with child's age. We identified that gestational age and birth weight were associated with cord serum lipids. Most of the lipid associations with gestational age were in the opposing direction to the associations with birth weight. Mode of birth was also associated with the cord serum lipidome, and the profile differed from that observed for gestational age and birth weight. There were marked changes in the ontogeny of the plasma lipidome from birth to 4 years of age: LC-PUFA and cholesteryl esters containing LC-P-UFAs were enriched in cord serum relative to the maternal serum. LC-PUFA containing lipid species altered between birth and 4 years with an enrichment in alkenylphosphatidylethanolamine species and a reduction in lysophospholipids and triglycerides. Concentrations of free LC-PUFAs decreased at 4

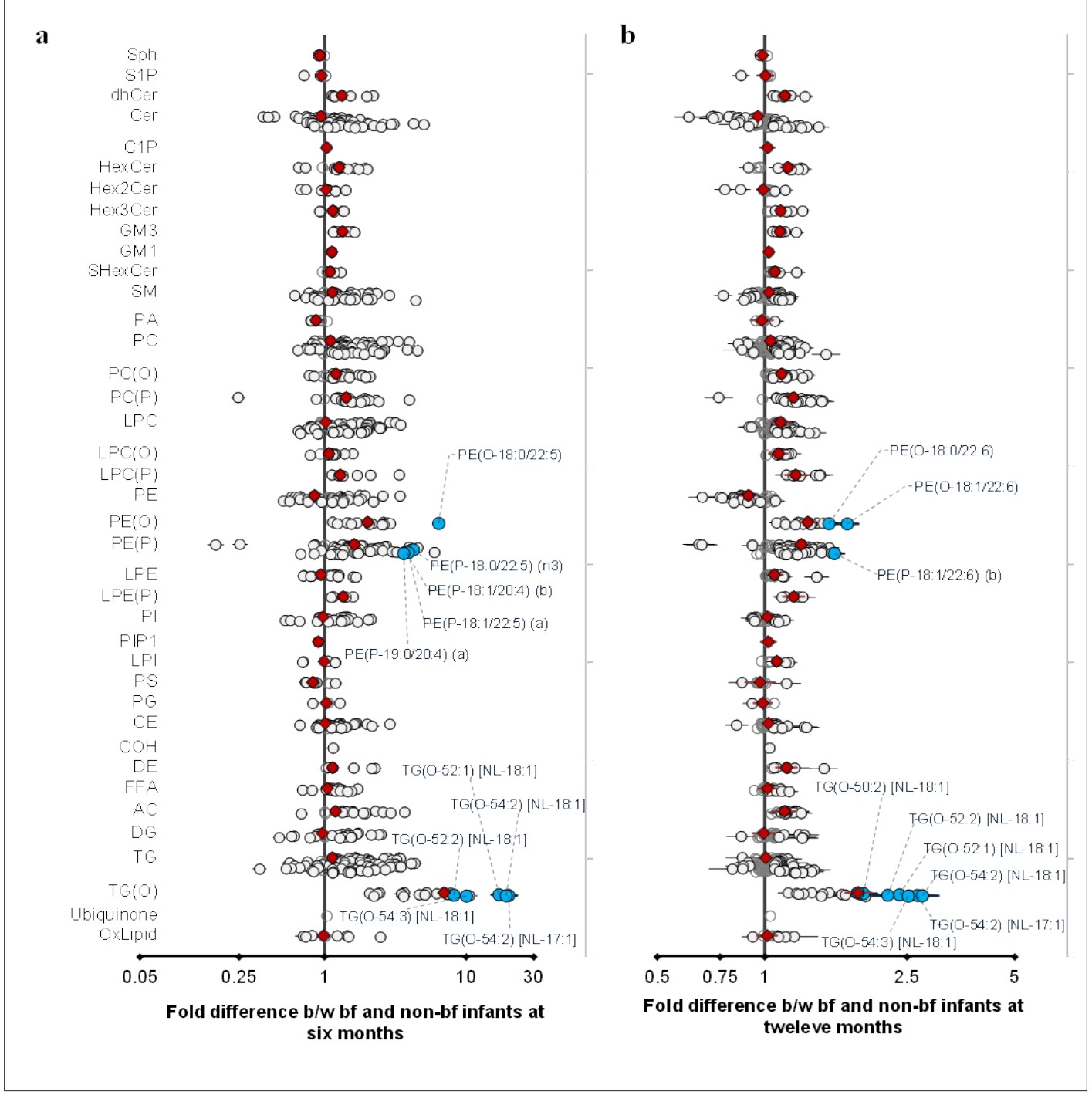

**Figure 6.** Breastfeeding influences the infant plasma lipidome. Linear regression of the lipidome on breastfeeding status at 6 and 12 months adjusting for age, sex, birth weight, gestational age (at 6 months), maternal pre-pregnancy BMI, gestational diabetes mellitus (GDM), maternal education, birth order. Results are shown as fold difference in lipids between breastfed and non-breastfed infants. (**a**) Lipid species associated with breastfeeding at 6 months. (**b**) Lipid species associated with breastfeeding at 12 months. Each circle on the plot represents a lipid species, grey open circles show non-significant lipid species (p > 0.05), white closed circles show significant lipid species (p < 0.05), the top 10 lipid species (10 lowest p-values) are shown in blue circles. Each red diamond represents a lipid class. All p-values were corrected for multiple comparisons. The horizontal bars (error bars) are shown for the significant lipid species (white closed circles), the top 10 lipid species (blue circles), Corrected p-value < $5.96 \times 10^{-189}$, $1.04 \times 10^{-38}$ respectively. The error bars represent 95% confidence intervals.

years, but the corresponding cholesteryl esters increased. Alkenylphosphatidylethanolamine species and alkyldiacylglycerol species were significantly elevated in breastfed infants at 6 and 12 months.

There has been a 30% increase in the incidence of late preterm births since 1980s, which now account for 75% of all preterm births. There is increasing epidemiological evidence suggesting infants born late preterm (34–36 weeks) and early term (37–38 weeks) are at higher risk of a host of adverse health outcomes including respiratory disorders, increased and prolonged hospitalisations, and developmental delay (*Crump et al., 2011*; *Consortium on Safe Labor et al., 2010*; *Talge et al., 2010*). On the other hand, the studies on association of gestational age with future health risk have been inconclusive. For example, Power et al. examined the associations between gestational age and blood pressure, HbA1c, clinical lipids, and BMI at 44–45 years of age in the 1958 British birth cohort. They identified that gestational age was inversely associated with blood pressure but there was no clear evidence of an association between gestational age and BMI or total cholesterol among men or between gestational age and HbA1c or LDL-cholesterol, HDL-cholesterol, and triglycerides in either sex (*Cooper et al., 2009*).

In our study, we found di- and triacylglycerols were positively associated with gestational age, which reflect an evolutionary adaptation in preparation for the metabolic stress of labour, a period of high energy expenditure (*Ghaemi et al., 2014*; *McCloskey et al., 2018a*). A concomitant increase in branched-chain fatty acids and corresponding phospholipids, a downstream product of the mitochondrial catabolism of branched-chain amino acids along with elevated levels of acylcarnitines, both suggest increased mitochondrial activity in babies with greater gestational age (*Bjørndal et al., 2018*).

Gestational age and birth weight showed overall inverse lipidomic profiles that were largely independent of each other. In infants with higher birth weight, the observed lower level of triglyceride levels could reflect better packaging within the adipose tissue vs. free availability in the circulation for energy mobilisation.

The decrease in cholesteryl esters and dehydrocholesterols suggest a decrease in cholesterol synthesis with increasing gestational age. This may relate to increased use of FFAs for energy production and so less for cholesteryl ester production but might also relate to a downregulation of the cholesterol biosynthesis pathway as the dehydrocholesterol is a precursor to cholesterol.

There were significant associations with multiple sphingolipid species, but overall these were very small, showing only 2–3% difference for a 10-day difference in gestational age. Thus while these lipids are known to associate with cardiometabolic outcomes in adults, the differences at birth cannot yet be ascribed to specific outcomes or risk. Further analyses will be required to define these relationships.

The lipid profile associated with gestational age resembles the lipid profile associated with BMI in adults. For example, in adults, acylcarnitines, di and triacylglycerols are positively associated with BMI and body weight (*Beyene et al., 2020*) like that seen in babies with higher gestational age. One possible explanation is that these lipid classes are involved in energy mobilisation and hence more abundant in circulation thereby acting as a source of FFAs which are required for various biological pathways (*Hamosh, 1987*; *Hellmuth et al., 2017*). Understanding the changes in lipid metabolism during the final stages of gestation may provide mechanistic insights into the increasingly recognised long-term health sequelae of preterm and late preterm births (*Kajantie et al., 2019*).

These results have also been observed in an ethnically diverse cohort, Growing Up in Singapore Towards healthy Outcomes (GUSTO), revealing that these lipidome changes associated with birth weight may be largely independent of race/ethnicity (*Mir et al., 2021*).

There are robust associations between mode of birth and child health outcomes, including risk of future overweight and obesity (*Wang et al., 2013*; *Li et al., 2013*). Elevated levels of di- and triacylglycerols and acylcarnitine in assisted VB, unscheduled CS and prolonged labour indicate that stress in utero and during birth increases energy mobilisation for high energy expenditure in these modes of birth (*Poznyak et al., 2020*; *Jung and Choi, 2014*; *McCloskey et al., 2018b*). However, a decrease in circulating FFAs and triacylglycerols in scheduled CS may also be a result of mothers often being prohibited from consuming food and drink before the procedure leading to a fasting state and so decreased availability of FFAs for transfer across the placenta. Additionally, mode of birth also affects the neonate's gut microbiota (*Reyman et al., 2019*). Newborns delivered vaginally show a higher diversity of bacteria than those born by CS (*Reyman et al., 2019*; *Francino, 2018*). The specific lipidome signature associated with each mode of birth may reflect the differences in the gut microbiome, although whether this could manifest at the time of delivery is uncertain. Considering data suggesting

an increased risk of metabolic disorders later in life associated with mode of birth (*Li et al., 2013*), it is important to understand differences in lipid metabolism by birth mode.

## Lipid differences in mother and newborn cord serum

Compared to the mother, we observed elevated levels of LC-PUFAs in cord serum which are critical for in utero brain development as well as the rapid brain development in the first year of life (*Innis, 2007*; *Koletzko et al., 2001*). The elevated levels of di- and triacylglycerols, and phospholipids observed in the maternal circulation most likely act as a source of fatty acids for the foetus. Lipoprotein lipase and endothelial lipase in the syncytiotrophoblast hydrolyse circulating maternal triacylglycerols and phospholipids respectively to provide the pool of fatty acids for the foetus (*Gil-Sánchez et al., 2012*). FFAs in the syncytiotrophoblast basal membrane are transferred to foetal circulation directly through facilitated diffusion or using fatty acid carriers (FAT/CD36) and fatty acid binding proteins (FABPpm, FATP4). In the foetal circulation, these fatty acids then bind to a fetoprotein and are transported to the liver, where they are re-esterified into complex lipids and transported into circulation (*Larqué et al., 2014*; *Shafrir and Khassis, 1982*).

LC-PUFA containing phosphatidylcholine species were also elevated in maternal serum. These phospholipids undergo an *sn*-1 cleavage by endothelial lipases in the syncytiotrophoblast producing LC-PUFA lysophosphatidylcholine species which are then transported across the placenta by the major facilitator superfamily domain containing 2A (MFSD2a) transporter proteins explaining the elevated levels of LC-PUFA containing LPCs in cord serum (*Ferchaud-Roucher et al., 2019*). Elevated levels of LC-PUFA containing cholesteryl esters in the cord serum might also be acting as an additional source of LC-PUFAs for the foetus.

Recent studies have shown altered placental function and thereby altered lipid transfer in mothers with gestational diabetes and higher pre-pregnancy BMI (*Prieto-Sánchez et al., 2017*; *Segura et al., 2017*). However, long-term ramifications of maternal metabolic conditions on offspring health are currently unknown. Our study contributes to the understanding of trans-placental transfer of specific lipid species. Further studies on how maternal metabolic conditions may impact this transfer may provide insight into perturbations in lipidome in utero and its long-term health consequences.

## Changes in lipid metabolism in the first 4 years of life

We observed pronounced changes in the lipid profiles in infancy and early childhood. Elevated levels of LC-PUFAs, cholesteryl esters, and lysophospholipids seen in the cord serum started to diminish as the child reached 6 months of age. However, at 12 months and 4 years of age, we observed an increase in odd chain, essential fatty acids and LC-PUFA containing phospholipids, which could be acting as a source of PUFA. Recent studies have shown LC-PUFA deficiency in several childhood disorders such as asthma, cystic fibrosis, attention-deficit/hyperactivity disorders, obesity, and diabetes (*Innis, 2007*; *Koletzko et al., 2001*). A balanced intake of omega-6 and omega-3 fatty acids is essential for homeostasis and normal development throughout the life cycle. A diet rich in omega-6 LC-PUFAs can shift the metabolism to an atherogenic/diabetic state (*Simopoulos, 2002*). On the other hand, the beneficial effects of omega-3 LC-PUFAs also depend on complex interaction between different nutrients, and polymorphism in genes involved in omega-3 fatty acid metabolism.

Clustering the lipids across all the time points revealed clear signatures of breastfeeding in clusters 2, 7, and 9 with lipid species of alkenylphosphatidylethanolamine and alkyldiacylglyerols showed remarkable increase at 6 months. Since the lipid clusters are designed as change in lipids over time, it is not possible to find how they are correlated with a particular outcome at one time point, rather we plan to investigate the association of these lipid clusters with growth trajectories, thereby enabling us to capture the lipid trajectories over time (manuscript in preparation).

Understanding changes in the lipidome early on in life provides a window of opportunity for early intervention and decrease the risk of future metabolic disorders.

It is well established that breast milk is rich in PUFAs. LC-PUFAs, especially arachidonic acid (20:4, n-6), eicosapentaenoic acid (20:5, n-3), and docosahexaenoic acid (22:6, n-3), have been shown to be critical for brain development and function in early life. While the actual mechanisms of how breastfeeding is associated with neurodevelopment remain unknown, LC-PUFAs in breast milk have been postulated to account for the effect of breastfeeding and improved cognition. We observed elevated levels of multiple lipid species containing these LC-PUFAs in breastfed infants at both 6 and

12 months. In addition, we observed a marked enrichment of alkenylphosphatidylethanolamine and alkyldiacylglycerol species in breastfed, relative to formula-fed, infants. Dysregulation of lipid metabolism is recognised as a primary driver of obesity and more recently of inflammation and immune regulation. Lipids make up 3% of the total breast milk composition. A small but important component of these lipids are alkylglycerols (AKGs), a class of ether lipids that constitute about 1% of total milk lipids. *Yu et al., 2019* recently reported that AKGs maintain beige adipose tissue (BeAT) in infants and delay the transformation of BeAT into white adipose tissue in mice, thereby protecting against obesity. They further report that breast milk AKGs are metabolised by adipose tissue macrophages to form platelet-activating factor, which ultimately activates IL-6/STAT3 signaling in adipocytes and triggers BeAT development in the infant. This study suggests that lack of AKG intake in infancy leads to premature loss of BeAT and increased fat accumulation and points to a role in immune cells in this process. Alkylglycerols can also be metabolised into alkyl- and alkenylphospholipids. Our group and others have identified that these classes of phospholipids are critical for human health and are depleted in obesity, diabetes, and cardiovascular disease (*Meikle and Summers, 2017*).

Breast milk represents the most abundant source of alkylglycerols in the human diet. While we see up to 17-fold increase in alkyldiacylglcyerols and alkenylphosphatidylethanolamine species at 6 and 12 months in breastfed infants, we did not observe any associations of breastfeeding at 12 months with 4 years circulating lipids. However, early modulation of these lipids might have long-term health effects, which is still to be understood.

## Effect of infant sex on lipids

There is evidence of sex differences in the lipidome from late gestation onwards (*Kelishadi et al., 2007*). For example, cord plasma concentrations of total cholesterol, HDL-cholesterol, and LDL-cholesterol are higher in girls than in boys (*Pac-Kożuchowska et al., 2018*), in keeping with our findings. We have previously identified sphingomyelin SM(d18:2/14:0) as the key lipid differentiating sex in adult cohorts (*Huynh et al., 2019*; *Beyene et al., 2020*). At birth, we did not find SM(d18:2/14:0) to be different between male and female babies. However, at age 4 years, SM (d18:2/14:0) was significantly associated with sex and was lower in boys relative to girls (10.42% lower in boys, $p = 5.53 \text{ E}^{-04}$). In adults, the differential activity of the fatty acid desaturase (FADS3) is responsible for the elevated levels of SM (d18:2/14:0) in women (*Beyene et al., 2020*; *Karsai et al., 2020*). Our results suggest that FADS3 starts exhibiting this differential activity by 4 years. Additionally, sex differences in circulating gonadotropin levels during the first few months of life is well documented (*Quigley, 2002*; *Shinkawa et al., 1983*; *Winter et al., 1975*). More recently, a differential hypothalamic-pituitary-adrenal axis activity in response to foetal glucocorticoid exposure has been recorded between sexes at birth (*Gifford and Reynolds, 2017*). Understanding the sex differences in the lipidome at birth and early life could provide mechanistic insights into sex differences in cardiometabolic diseases.

## Strengths and limitations

Major strengths of this study include repeated and extensive sampling in a large population-derived cohort at in pregnancy and early life with high quality, standardised metadata.

The BIS cohort provides an opportunity to study the effects of environmental factors during early infancy. However, the lack of generalisability (less ethnically diverse, more affluent than the general population) may limit the understanding of outcomes in different ethnicities and socio-economic strata. It would be valuable to follow the children through adolescence to understand the long-term modulation of lipid metabolism and early markers of future health or disease outcomes. We are currently investigating these results in another large birth cohort, based in Singapore – the GUSTO cohort to investigate whether similar changes are evident in a population of Asian ethnicity. In this study, plasma lipidomics was performed on serum (maternal and newborn time points) and plasma (6, 12 months, and 4 years). Several studies have pointed out to minor differences in lipid levels of these two matrices, however, these differences do not show any associations with biological functions (*Lagarde et al., 2010*).

## Conclusion

We report in-depth lipidomic profiling in both gestation and the earliest stages of life, thereby providing insight into the ontogeny of lipid metabolism through infancy and early childhood. Despite

the limitations, the extensive lipidomic profiling gives us insight into perinatal and postnatal factors associated with lipid metabolism during the first 4 years of life. There is now growing evidence of foetal programming emphasising the profound and sustained impact of intrauterine and early life factors to foetal health and development of metabolic diseases in later life. Understanding the baseline characteristics of lipid metabolism at birth and throughout early years provide a resource for further studies to elucidate the clinical implications. The datasets resulting from this study now provide a rich resource to further investigate the relationship between lipid metabolism and health outcomes during early life. Children within this study are continuing to be followed and future outcome data will build on these initial early life findings.

## Acknowledgements

The establishment work and infrastructure for the BIS was provided by the Murdoch Children's Research Institute, Deakin University, and Barwon Health. Funding has been provided by the National Health and Medical Research Council of Australia (607370, 1009044, 102997, 1082037, 1076667, and 1084017), the Jack Brockhoff Foundation, the Scobie Trust, the Shane O'Brien Memorial Asthma Foundation, the Our Women's Our Children's Fundraising Committee Barwon Health, the Rotary Club of Geelong, the Shepherd Foundation, the Ilhan Foundation, and the Operational Infrastructure Support Program of the Victorian Government. We acknowledge the participation and commitment of all the families in the BIS.

## Additional information

### Group author details

**Barwon Infant Study Investigator team**
**Peter Vuillermin; Fiona Collier; Anne-Louise Ponsonby; John Carlin; Katie Allen; Mimi Tang; Richard Saffery; Sarath Ranganathan; David Burgner; Terry Dwyer; Peter Sly**

### Funding

| Funder | Grant reference number | Author |
|---|---|---|
| National Health and Medical Research Council | A*STAR-NHMRC joint call funding (1711624031). | Peter Meikle<br>Markus R Wenk<br>Neerja Karnani<br>Fiona Collier<br>Richard Saffery<br>Peter Vuillermin<br>Anne-Louise Ponsonby<br>David Burgner |

The funders had no role in study design, data collection and interpretation, or the decision to submit the work for publication.

### Author contributions

Satvika Burugupalli, Formal analysis, Methodology, Visualization, Writing – original draft, Writing – review and editing; Adam Alexander T Smith, Formal analysis, Writing – review and editing; Gavriel Oshlensky, Tingting Wang, Alexandra George, Anh Nguyen, Natalie Mellett, Michelle Cinel, Methodology; Kevin Huynh, Corey Giles, Sudip Paul, Thy Duong, Methodology, Writing – review and editing; Sartaj Ahmad Mir, Li Chen, Validation, Writing – review and editing; Markus R Wenk, Conceptualization, Funding acquisition, Methodology, Writing – review and editing; Neerja Karnani, Funding acquisition, Validation, Writing – review and editing; Fiona Collier, Project administration, Writing – review and editing; Richard Saffery, Supervision, Writing – review and editing; Peter Vuillermin, Anne-Louise Ponsonby, David Burgner, Funding acquisition, Project administration, Supervision, Writing – review and editing; Peter Meikle, Conceptualization, Funding acquisition, Investigation, Methodology, Supervision, Writing – review and editing; Barwon Infant Study Investigator team, Conceptualization, Data curation, Project administration

## Author ORCIDs

Satvika Burugupalli ⓘ http://orcid.org/0000-0002-8866-1756
Alexandra George ⓘ http://orcid.org/0000-0001-7079-6647
Fiona Collier ⓘ http://orcid.org/0000-0002-5438-480X
Anne-Louise Ponsonby ⓘ http://orcid.org/0000-0002-6581-3657
David Burgner ⓘ http://orcid.org/0000-0002-8304-4302
Peter Meikle ⓘ http://orcid.org/0000-0002-2593-4665

## Ethics

Human subjects: The original study was granted by the Barwon Health Human Research and Ethics Committee (HREC 10/24).

## Decision letter and Author response

Decision letter https://doi.org/10.7554/eLife.72779.sa1
Author response https://doi.org/10.7554/eLife.72779.sa2

## Additional files

### Supplementary files

• Supplementary file 1. Supplemetary data for the analyses.
• Transparent reporting form

### Data availability

Due to the consent obtained during recruitment process, it is not possible to make all data publicly available. Access to BIS data including all data used in this paper may be requested through the BIS Steering Committee by contacting the corresponding author. Requests to access cohort data are considered on scientific and ethical grounds and, if approved, provided under collaborative research agreements. Deidentified cohort data can be provided in Stata or CSV format. All statistical methods used are referenced within the methods section. Data that is not subject to data-sharing restrictions can be found in Supplementary File 1. Additional project information, including cohort data description and access procedures, is available at the project's website https://www.barwoninfantstudy.org.au/.

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
