## [Editor Report]

The paper is well written and the experimental approach is sound. It contains a large amount of data and addresses an important and an understudied area of science. The authors were very responsive to the reviewers' comments and the reviewers unanimously decided that the paper is significantly improved and is ready for publication. The paper will likely have a high impact on the field, as it identifies changes in plasma lipids at the early stages after birth and its link with factors related to pregnancy.

---

## [Decision Letter]

**Decision letter after peer review:**

Thank you for submitting your article "Ontogeny of plasma lipid metabolism in pregnancy and early childhood: a longitudinal population study" for consideration by *eLife*. Your article has been reviewed by 3 peer reviewers, including Hossein Ardehali as Reviewing Editor and Reviewer #1, and the evaluation has been overseen by Y M Dennis Lo as the Senior Editor. The following individual involved in review of your submission has agreed to reveal their identity: Liang Wei Wang (Reviewer #3).

Essential revisions:

The paper has been discussed extensively among the reviewers. The Reviewers found the paper interesting and identified significant merit in the work. However, several major weaknesses were also identified that reduced enthusiasm and prevented the publication of the work in the present form. The main shortcomings of the work are mainly the lack of deep analysis of the data and also presentation of the results. It is important for the author to include the following studies in the revised version:

1) Better definition of the significance of the findings, as requested by Reviewers #1 and #3.

2) Better analysis of the data that employ other methods besides linear regression analysis to assess high-risk, medium-risk, and low-risk clusters of lipid species.

3) Discussion on how various complications e.g., breech births, meconium aspiration, might associate with gestational lipid signatures, with immediate implications for triage and emergency planning;

4) Inclusion of discussion on how milk as a source of lipids can influence the ontogeny of the child's lipidome;

5) Limiting the paper and the results to lipid data only during labor, delivery and the first year of life.

*Reviewer #1 (Recommendations for the authors):*

This is a well written paper and a large amount of data are provided. I have the following comments:

1. Figures 2-4 are very difficult to understand. What do each point in the blot represent? What does the line around the data points represent? It is important the authors provide more explanation for readers who do not have familiarity with these plots.

2. There are a number of lipids that may have toxic effects on certain cells. For example ceramide has been shown to have cardiotoxic effects in adults. The authors show that ceramide levels change, but do not discuss what the biological significance of these findings are. In general, although the paper provides a large amount of data, it lacks a clear explanation for what these changes mean. The authors need to go beyond simply showing their lipidomic data and provide more physiological relevance to their findings.

3. Additional environmental variables and their link to changes in childhood serum lipids need to be studied and included in the paper.

*Reviewer #2 (Recommendations for the authors):*

This is an important and well-written manuscript. While the work is largely descriptive, the authors provide important information on the natural biological variation of the circulating lipidome from gestation to up to 4 years of age, along with corresponding data on their mothers. However, there are major concerns:

1. The authors show with regression analyses that certain lipid species are increased/decreased in response birth and year 4 BMI and weight. However, it is highly unlikely these individual lipid species are acting in isolation. It is more likely that there are various high-risk, moderate-risk, and low-risk "clusters" of lipid species that change over time with respect to each other which are associated with birth weight and ultimately year 4 BMI. Linear regression is not sufficient to resolve such clusters, and time-series principal component analysis should be used (eg, PMID 32504647)

2. The significance of the work would be significantly strengthened if the authors performed machine learning analysis of the dynamic lipidome to identify particular species associated with birth weight and year 4 BMI to complement traditional statistical approaches.

*Reviewer #3 (Recommendations for the authors):*

Cohort description:

The description of the cohort can be further improved. Although the study cohort's demographic details have been published in reference 12, those numbers are woefully outdated. It is not clear how the new subjects recruited in the past 6 years have altered the demographic structure of the cohort. Additionally, more pertinent details about the cohort in relation to metabolism should be presented in Supplementary Table 1 e.g., caloric and fat intake during pregnancy, breast milk lipid content.

What is the reason for excluding the 1-, 9- and 24-month time points?

Language:

The language of the manuscript can be changed to improve clarity and ease of reading.

– The authors use American and British English words interchangeably e.g., "fetal" and "foetal". I suggest standardization.

– For example, "postnatal age" can be substituted with "child's age".

– In the Statistical Analysis section, it is not clear who "participant" refers to in paragraph 2.

– The manuscript title is somewhat misleading – sera, not plasma, were collected at the gestational and birth time points. Hence, the title should read "blood lipid metabolism", instead of "plasma lipid metabolism". Consequently, the Methods section should include a line on this difference in sample type for avoidance of doubt.

– The authors noted a similarity between the β coefficients of lipid species associated with BMI at 4 years and pre-pregnancy BMI in mothers (Supplementary Figure 6b). However, the authors should qualify this as a very weak correlation.

– On page 20, the authors say that the lipid profiles of 4-year-old children associated with BMI closely resembled that of adults. It is not clear whether this said adult cohort is the one referred to by reference 8. Even so, the data stated in support of the authors' claims are found in Supplementary Figures5 and 6 and, Supplementary Tables 13-16. However, none of these data actually provide substantive support, since a PCA is the best first step to establishing that fact. On a related note, it will also be quite interesting to find out how the lipidome differs between healthy non-pregnant females in reference 8's cohort and healthy pregnant women in this study.

– On page 22, the authors order assisted VB, non-assisted VB, unscheduled CS and a 4th CS. I believe the word, "schedule", is missing here. In the same paragraph, the authors say that [the] "mode of birth also effects the neonate's gut microbiota". Here, "affects" should be used in place of "effects".

– On page 23, paragraph 2, there is a missing "as" in line 5. On page 24, the word, "the", is missing in paragraph 2 line 13 and paragraph 3 line 1.

– On page 25, the authors assert that FADS3 "starts exhibiting this differential activity at four years". This should be rephrased as "by four years" since the activity difference could have happened any time between the prior time point and 4 years of age.

Presentation:

The presentation of figures needs to be improved. For example, in the Results section, Figures1b and 1c are referenced after Figure 1d in paragraph 2.

In Figure 1b, cord sera are represented as a distinct cluster to other child-derived samples. How do the authors know that this represents a real biological difference that is not due to a difference in sample processing? Figure 1c should be re-plotted with PC3 to achieve better separation.

Page 11 presumably refers to Figure 2. But no textual reference is actually made.

Figures2 to 4 should have category labels on the y-axis to aid reading. Lipid species are classified by molecular similarity. However, it will be much more intuitive and useful to categorize them according to pathway relatedness or molecular function. If possible, dendrograms should also be drawn to indicate molecular relatedness.

Supplementary Figure 2 is very ambiguous – it is not clear to me how certain lipid species have been highlighted/cherry-picked. The authors should define clear cut-offs.

Figure legends can be improved for clarity. For example, for p-values, 10-15 should be used, rather than E-15. Red diamonds are said to represent lipid classes – but what does this mean exactly? Is it the mean or median value of all species within the class? In Supplementary Figure 5, the legend title says, "Association of infant BMI with lipid species". However, panel d relates to 4-year-old children i.e., toddlers, rather than infants; the correct word to use is "child".

Other Suggestions and Points Requiring Clarification:

The authors note the interesting observation that acylcarnitine, DAGs and TAGs were more abundant with longer duration of labor, alluding to a role for fueling the strenuous activity of labor. How does one know that the increased levels of these lipid species are due to the physiology of labor per se (on page 22, the authors write, "stress in utero and during birth increases energy mobilization for high energy expenditure in these modes of birth"), rather than a result of mothers often being prohibited from consuming food and drink (essentially an enforced state of fasting) during labor?

A few clinical issues related to breastfeeding and child lipidomic changes could have been but were also not addressed by the authors using the present dataset. For example, it would be important to know whether the mode of feeding – exclusive breastfeeding or mixed feeding – influences the development of the child's lipidome. Other questions that can be answered as well include: (a) delineating the influences of different types of formula milk (cow versus goat, for example) on the child's lipidome at different time points, (b) definition of particular breast milk lipid signatures that correlate to higher infant/toddler BMI, and (c) determination of breast milk lipid content and compositions for GDM and healthy mothers.

---

## [Author Response]

Essential revisions:The paper has been discussed extensively among the reviewers. The Reviewers found the paper interesting and identified significant merit in the work. However, several major weaknesses were also identified that reduced enthusiasm and prevented the publication of the work in the present form. The main shortcomings of the work are mainly the lack of deep analysis of the data and also presentation of the results. It is important for the author to include the following studies in the revised version:1) Better definition of the significance of the findings, as requested by Reviewers #1 and #3.

While there are large population studies that have established dysregulation of lipid metabolism to be associated with metabolic syndrome in adults, very little is known about lipid metabolism in utero and in early life, nor about the ante-natal factors affecting lipid metabolism in infants and how this manifest in later life or whether these changes persist. This is partly due to the long follow up times required for such studies in order to investigate clinical outcomes in later life. Additionally, in depth lipidomic profiling data of infants and young children are scarse. Accordingly, we have made inferences from adult studies where possible and made changes to the discussion explaining the potential implications of early metabolic changes to cholesteryl esters and sphingolipids. These changes are added to the discussion in the manuscript

Discussion

“There has been an 30% increase in the incidence of late preterm births (LPIs) since 1980s. which now account for 75% of all pre-term births. […] Understanding the changes in lipid metabolism during the final stages of gestation may provide mechanistic insights into the increasingly recognised long-term health sequelae of preterm and late preterm births [11].”

2) Better analysis of the data that employ other methods besides linear regression analysis to assess high-risk, medium-risk, and low-risk clusters of lipid species.

In addition to the existing analyses to study the changes in lipidome over the first four years, we performed time-series clustering to capture the changes of all the lipid species across all the time points. We utilised Dynamic Time Warping (DTW) distance as a dissimilarity measurement (Aghabozorgi, S. et.al. Time-series clustering – A decade review. Information Systems, 2015). We used the partitional clustering method with partitions around medoids (PAM) to cluster the lipid species. We identified 10 different lipid clusters, each of them showing unique lipid trajectories over the first four years of life (Figure 5 —figure supplement 2, Supplementary Table 10).

The results are now included in page 22, line 22 – page 24, line 14 and discussed in page 33, line 19.

“Additionally, we performed time-series clustering to capture the changes of all the lipid species across all the time points. […] Acylcarnitine and fatty acid species, that were elevated in cord serum, compared to maternal serum, increased until 6 months and later decreased at 12 months and further at 4 years suggesting a change in energy metabolism over the first 4 years.”

3) Discussion on how various complications e.g., breech births, meconium aspiration, might associate with gestational lipid signatures, with immediate implications for triage and emergency planning:

In this study, we have studied four different modes of birth: Unassisted vaginal delivery (n = 525), assisted vaginal delivery (vacuum assisted + forceps assisted) (n = 214), scheduled cesarean (n = 178), and unscheduled cesarean births (n = 155). The dataset does not include further classification of unscheduled cesarean births. While it may be informative to have such information and evaluate lipid signatures associated with such complications, we suspect that the low numbers of such cases within a cohort would be underpowered for such analyses. Additionally, they are all, to varying degrees, stress-inducing for the foetus and if they are associated with any changes, they could probably show a similar profile.

We have now included this explanation in the Results section (page 17, line 6 -11)

“In this study, we compared four different birth modes: assisted vaginal birth (assisted VB, n = 214) (including both forceps- and vacuum-assisted birth), scheduled cesarean (CS, n = 178) and unscheduled cesarean (unscheduled CS, n = 155), with unassisted vaginal birth (VB, n = 525) as reference. Unscheduled CS accounted for all the emergency CS including breech births, while individual data on the reason for the emergency was not recorded.”

4) Inclusion of discussion on how milk as a source of lipids can influence the ontogeny of the child's lipidome;

We have now included an analysis of the association of breast feeding with plasma lipids at 6 and 12 months of age and observed strong associations with most lipid species and classes.

This has been included in the results page 26 line 1-15, Figure 6a and 6b, Supplementary Tables 13 and 14. We further discuss how breastfeeding can influence the lipid metabolism in the first year of life page 34, line 3 to page 35, line 7.

“Breastfeeding impacts lipid metabolism in the first year of life. We investigated the association of breastfeeding with plasma lipids at 6 months and 12 months. Of the 776 lipid species 664 lipid species (90%) were significantly associated with breastfeeding at 6 months, and 438 (of 773, 65%) were significantly associated at 12 months (Supplementary tables 13 and 14). Of particular note, species of alkyl and alkenylphospholipids (plasmalogens) and alkyldiacylglycerides (TG-O) were markedly elevated in breastfed infants. At a class level these elevations were of the order of 2-4 fold, while individual species of alkydiacylglycerols: TG(O-54:2) [NL-17:1], TG(O-54:2) [NL-18:1], TG(O-52:1) [NL-18:1] were elevated 17 – 19 fold at 6 months. However, at 12 months, this effect size was only 2-4 fold, given the introduction of a wide variety of foods along with breast milk (Figure 6). In addition to the dramatic increase in these ether lipid species, we also observed an increase in many species containing odd- and branched-chain fatty acids such as phosphatidylcholine PC(33:0), sphingomyelin SM(d19:1/24:1) and others. These odd- and branched-chain fatty acids represent a novel, potentially bioactive, class of lipids enriched in breast fed infants.”

5) limiting the paper and the results to lipid data only during labor, delivery and the first year of life.

Based on the suggestions from the editors, we have now limited the manuscript to analyses of the circulating lipids in the first year of life. Additionally, we extended into the study of changes in lipidome over the first four years of life. Based on the suggestions from reviewer 2, we performed time-series clustering to capture the changes of all the lipid species across all the time points. We utilised Dynamic Time Warping (DTW) distance as a dissimilarity measurement. We used the partitional clustering method with partitions around medoids (PAM) to cluster the lipid species. We identified 10 different lipid clusters, each of them showing unique lipid trajectories over the first four years of life (Figure 5 —figure supplement 2, Supplementary Table 10).

We have also included the analyses on association of breastfeeding on infant lipids in the first year of life (Figure 6, Supplementary Table 13, 14).

The manuscript now includes the following analyses:

1. Analyses of association of gestational age and birthweight with newborn cord serum lipids.

2. Analyses of association of mode of birth and duration of labour with the newborn cord serum lipids.

3. Enrichment analyses of lipid species in cord serum relative to maternal plasma.

4. Analyses of changes in the circulating lipidome over the first four years.

5. Analyses of association of sex with circulating lipids at all the time points.

6. Analyses of the association of plasma lipids at 6 months and 12 months with breastfeeding in the first year of life.

Reviewer #1 (Recommendations for the authors):This is a well written paper and a large amount of data are provided. I have the following comments:1. Figures 2-4 are very difficult to understand. What do each point in the blot represent? What does the line around the data points represent? It is important the authors provide more explanation for readers who do not have familiarity with these plots.

The authors agree. In addition to the Figure legends, we have now included a section “data representation” under the methods section (Page 10, line 9-15)

“Data Representation: The results from the association analyses are presented as forest plots in Figures 2-4 and 6 where each lipid species (or class) is represented by a marker showing either a positive association (increases with an increase in the outcome) or a negative association (decreases with an increase in the outcome). The colour of the markers relate to the significance of the association as indicated in the figure legends. The lipid species are grouped into lipid class and subclass and ordered into categories (sphingolipids, phospholipids and other lipids) each separated by a grey dotted line.”

2. There are a number of lipids that may have toxic effects on certain cells. For example ceramide has been shown to have cardiotoxic effects in adults. The authors show that ceramide levels change, but do not discuss what the biological significance of these findings are. In general, although the paper provides a large amount of data, it lacks a clear explanation for what these changes mean. The authors need to go beyond simply showing their lipidomic data and provide more physiological relevance to their findings.

The authors agree with the concerns raised by the reviewer. While there are large population studies that have established dysregulation of lipid metabolism to be associated with metabolic syndrome, very little is known about the lipid metabolism in early life and the in-utero and ante-natal factors affecting lipid metabolism in infants and how its manifested in later life. This is partly, due to the long follow up times required for such studies and most of them have containing sufficient variation in gestational age have begun to reach adulthood allowing other potential outcomes to be investigated. Additionally, in depth lipidomic profiling of infants and young children is only currently emerging and our study is one of such early ones. We have tried our best to bring in inference from the adult population where possible. Accordingly, changes have been made to the discussion explaining changes to cholesteryl esters and sphingolipids additionally.

Please refer to Discussion (page 29, line 17 to page 31, line 8)

Refer to essential revisions 1

3. Additional environmental variables and their link to changes in childhood serum lipids need to be studied and included in the paper.

The authors recognise that several environmental variables such as exposure to phthalates are known to be associated with neurodevelopment. However, the manuscript as such presents a lot of data, and is now scaled back based on the recommendations from the editors. A separate study by our collaborators Prof. Anne Louise Ponsonby et al., focusses on understanding the effect of environmental factors on lipid metabolism at birth. In this manuscript we focussed on understanding the baseline lipid metabolism in newborns, infants and children at four years of age. We have made attempts to understand the association of antenatal factors with lipid metabolism at birth. The overall goal of this manuscript and the resulting dataset is to provide a framework for future studies to understand and identify several factors in early life that can cause lipid dysregulation.

We have now clarified this in the abstract and introduction.

Abstract (Page 2, line 7-13)

“We studied the effect of antenatal factors such as gestational age, birth weight and mode of birth on lipid metabolism at birth; changes in the circulating lipidome in the first four years of life and the effect of breastfeeding in the first year of life. From this study, we aim to generate a framework for deeper understanding into factors effecting lipid metabolism in early life, to provide early interventions for those at risk of developing metabolic disorders including cardiovascular diseases”

Introduction (Page 4, line 14 – 21)

“This manuscript emphasises on understanding the lipid metabolism in newborns and infants in the first year of life, specially focusing on the association of cord serum lipids with antenatal factors: gestational age, birthweight, mode of birth and duration of labour. We also establish the association of breastfeeding with infant plasma lipids at 6 months and 12 months of age. We provide a baseline characterisation of circulating lipids in the mother and newborn and changes in the circulating plasma lipids from birth to four years. In summary, the results from this study provide a framework for future studies/ analyses to understand lipid metabolism in early life and provide early interventions for lipid dysregulation.”

Reviewer #2 (Recommendations for the authors):This is an important and well-written manuscript. While the work is largely descriptive, the authors provide important information on the natural biological variation of the circulating lipidome from gestation to up to 4 years of age, along with corresponding data on their mothers. However, there are major concerns:1. The authors show with regression analyses that certain lipid species are increased/decreased in response birth and year 4 BMI and weight. However, it is highly unlikely these individual lipid species are acting in isolation. It is more likely that there are various high-risk, moderate-risk, and low-risk "clusters" of lipid species that change over time with respect to each other which are associated with birth weight and ultimately year 4 BMI. Linear regression is not sufficient to resolve such clusters, and time-series principal component analysis should be used (eg, PMID 32504647)

Upon the recommendation from the editors, we have now excluded the analyses of association of lipids with BMI. Based on reviewer 2’s comments, we performed time-series clustering to capture the changes of all the lipid species across all the time points. We utilised Dynamic Time Warping (DTW) distance as a dissimilarity measurement. We used the partitional clustering method with partitions around medoids (PAM) to cluster the lipid species. We identified 10 different lipid clusters, each of them showing unique lipid trajectories over the first four years of life (Figure 5 – Supplementary figure 2, Supplementary Table 10).

The results are included in page 22, line 22 – page 24, line 12 and discussed page 33, line 18

Refer to essential revisions 2

2. The significance of the work would be significantly strengthened if the authors performed machine learning analysis of the dynamic lipidome to identify particular species associated with birth weight and year 4 BMI to complement traditional statistical approaches.

The authors agree that analysis of the dynamic lipidome is required. We also recognise the limitations of the use of BMI as a measure for obesity in children. A child’s weight status is different from adult BMI categories. Children’s body composition varies as they age and varies between boys and girls. Therefore, BMI levels among children and teens need to be expressed relative to other children of the same age and sex. Hence, growth trajectories developed using BMI are widely used to track child’s growth 9 (Wickramasinghe, V.P., et al., *Validity of BMI as a measure of obesity in Australian white Caucasian and Australian Sri Lankan children.* Annals of Human Biology, 2005.; Himes, J.H., *Challenges of accurately measuring and using BMI and other indicators of obesity in children.* Pediatrics, 2009)

We are currently performing analyses of growth trajectories and will then be able to apply machine learning approaches to identify changes to the lipidome associated with adverse growth trajectories. However, these analyses are outside the scope of this manuscript. Here, we seek to provide an overview of the data, the major associations, and areas for future exploration. We trust the reviewer recognises the complexity and richness of the data and our decision not to present all analyses in a single manuscript

Reviewer #3 (Recommendations for the authors):Cohort description:The description of the cohort can be further improved. Although the study cohort's demographic details have been published in reference 12, those numbers are woefully outdated. It is not clear how the new subjects recruited in the past 6 years have altered the demographic structure of the cohort. Additionally, more pertinent details about the cohort in relation to metabolism should be presented in Supplementary Table 1 e.g., caloric and fat intake during pregnancy, breast milk lipid content.

The authors have now made necessary changes to the cohort description. Recruitment was limited to 2010-13, so there has been no change in baseline demographics over the ensuing years. This has now been clarified (page 5, line 6-7)

We have now added additional details in Supplementary Table (S1) including

– Pre-pregnancy BMI

– Gestational weight gain

– Dietary caloric and fat intake for the mothers, and

– Duration of breastfeeding for the children

What is the reason for excluding the 1-, 9- and 24-month time points?

Plasma/ serum samples at these time points are not available.

Clarified under the Research design and cohort in the Methods section (Page 4, line 6-7):

“Maternal serum samples were collected at 28 weeks gestation. Child serum/ plasma samples were available only at birth, 6 months, 12 months and 4 years.”

Language:The language of the manuscript can be changed to improve clarity and ease of reading.– The authors use American and British English words interchangeably e.g., "fetal" and "foetal". I suggest standardization.

This is now corrected

– For example, "postnatal age" can be substituted with "child's age".

This is now corrected

– In the Statistical Analysis section, it is not clear who "participant" refers to in paragraph 2.

This is now corrected.

Page 9, line 12

“Associations between maternal/ infant/ children characteristics and lipid species were determined using multiple linear regression, adjusting for appropriate covariates in each analysis”

– The manuscript title is somewhat misleading – sera, not plasma, were collected at the gestational and birth time points. Hence, the title should read "blood lipid metabolism", instead of "plasma lipid metabolism". Consequently, the Methods section should include a line on this difference in sample type for avoidance of doubt.

The authors agree to the reviewer’s comments. However, blood lipid metabolism could also be misleading as we have utilised serum and plasma, derived from blood, rather than whole blood.

Hence, we have changed the title to “Ontogeny of circulating lipid metabolism in pregnancy and early childhood: a longitudinal population study”.

We have also clarified this in the methods section (Page 7, line 20 – page 8, line 2)

“Any lipid differences based on the matrix must be kept in mind as serum samples were used during gestation, and at birth vs plasma samples at other time points. Several studies have reported matrix-associated differences in species of lysophospholipids, diacylglycerols, free long chain fatty acids, and oxidised fatty acids. However, these differences are unlikely to reflect biological processes in the body. These differences could be reflected in change in lipid levels observed at birth and 6 months, but not affect the association with outcomes.”

– The authors noted a similarity between the β coefficients of lipid species associated with BMI at 4 years and pre-pregnancy BMI in mothers (Supplementary Figure 6b). However, the authors should qualify this as a very weak correlation.

At the request of the editor, we have now excluded the analyses.

– On page 20, the authors say that the lipid profiles of 4-year-old children associated with BMI closely resembled that of adults. It is not clear whether this said adult cohort is the one referred to by reference 8. Even so, the data stated in support of the authors' claims are found in Supplementary Figures5 and 6 and, Supplementary Tables 13-16. However, none of these data actually provide substantive support, since a PCA is the best first step to establishing that fact. On a related note, it will also be quite interesting to find out how the lipidome differs between healthy non-pregnant females in reference 8's cohort and healthy pregnant women in this study.

At the request of the editor, we have now excluded the analyses.

A similar analysis has been performed in the Growing up in Singapore towards healthy outcomes (GUSTO) cohort, where they have followed up the pregnant women in the cohort 6 years post pregnancy. In the analyses, they have compared the lipid profiles associated with BMI in pregnant women to the lipid profiles associated with BMI in the same women, 6 years post pregnancy (Ref: Mir, S.A., et al., *Developmental and Intergenerational Landscape of Human Circulatory Lipidome and its Association with Obesity Risk.* bioRxiv, 2021). However, in the BIS cohort, we don’t have the follow up samples available.

– On page 22, the authors order assisted VB, non-assisted VB, unscheduled CS and a 4th CS. I believe the word, "schedule", is missing here.

This sentence have now been deleted (Page 31, line 12)

In the same paragraph, the authors say that [the] "mode of birth also effects the neonate's gut microbiota". Here, "affects" should be used in place of "effects".

This is now corrected (Page 31, line 19)

Additionally, mode of birth also affects the neonate’s gut microbiota.

– On page 23, paragraph 2, there is a missing "as" in line 5. On page 24, the word, "the", is missing in paragraph 2 line 13 and paragraph 3 line 1.

These are now corrected

Page 32, line 7

“The elevated levels of di- and tri-acylglycerols, and phospholipids observed in the maternal circulation most likely act as a source of fatty acids for the foetus.”

Page 34, line 1

“Understanding changes in the lipidome early on in life provides a window of opportunity for early intervention and decrease the risk of future metabolic disorders”

Page 35, line 8

“There is evidence of sex-differences in the lipidome from late gestation onwards…”

– On page 25, the authors assert that FADS3 "starts exhibiting this differential activity at four years". This should be rephrased as "by four years" since the activity difference could have happened any time between the prior time point and 4 years of age.

This is now corrected (Page 35, line 17)

“Our results suggest that FADS3 starts exhibiting this differential activity by four years.”

Presentation:The presentation of figures needs to be improved. For example, in the Results section, Figures1b and 1c are referenced after Figure 1d in paragraph 2.

This is now corrected (Page 11, line 1-7)

“Overall lipid levels increased with age: maternal serum had higher total lipid levels than infant plasma, which in turn had higher levels than cord serum (Figure 1b). Principal Component Analysis (PCA) of the lipidomic data from all participants at all the time points revealed a clear separation of maternal, newborn, and infant samples across the first and second principal components (Figure 1c). A PCA of the infant samples showed further separation of the 6, 12- and 4-year time points (Figure 1d, 1e).”

In Figure 1b, cord sera are represented as a distinct cluster to other child-derived samples. How do the authors know that this represents a real biological difference that is not due to a difference in sample processing?

The authors acknowledge that samples at birth and other child time points are different matrices, serum and plasma.

We have also clarified this in the methods section (Page 7, line 20 – page 8, line 2)

“Any lipid differences based on the matrix must be kept in mind as serum samples were used during gestation, and at birth vs plasma samples at other time points. Several studies have reported matrix-associated differences in species of lysophospholipids, diacylglycerols, free long chain fatty acids, and oxidised fatty acids. However, these differences are unlikely to reflect biological processes in the body. These differences could be reflected in change in lipid levels observed at birth and 6 months, but not affect the association with outcomes”

Figure 1c should be re-plotted with PC3 to achieve better separation.

The authors have now included a PC2 vs PC3 plot (Figure 1e) in Figure 1

Page 11 presumably refers to Figure 2. But no textual reference is actually made.

This is now corrected (page 15, line 3)

461 (of 733, 63%) cord lipid species were associated with gestational age and 299 (of 733, 41%) lipid species were associated with birth weight (Figure 2a, 2b).

Figures 2 to 4 should have category labels on the y-axis to aid reading. Lipid species are classified by molecular similarity. However, it will be much more intuitive and useful to categorize them according to pathway relatedness or molecular function. If possible, dendrograms should also be drawn to indicate molecular relatedness.

The lipid species are arranged according to the pathway relatedness. All the ceramides and sphingomyelins are under one section, followed by all phospholipids and lyso-phospholipids under one section, cholesterols, triglycerides and diacylglycerides are under one section, each divided by a grey dotted line. We have also included a supplementary figure 1, to interpret the lipid pathways. (Page 42, Supplementary Figure 1)

This information has now been included in methods (page 10, line 12-14) to aid better understanding.

“The lipid species are grouped into lipid class and subclass and ordered into categories (sphingolipids, phospholipids and other lipids) each separated by a grey dotted line.”

Supplementary Figure 2 is very ambiguous – it is not clear to me how certain lipid species have been highlighted/cherry-picked. The authors should define clear cut-offs.

The phospholipids associated with gestational age and those associated with birth weight showed clear opposing trends. We chose odd chain containing phospholipids to highlight that difference in directionality.

Modified the figure legend (Page 44, Figure 2 —figure supplement 2):

Supplementary Figure 3 a, Volcano plot showing newborn phospholipids associated with gestational age. b, Volcano plot showing newborn plasma phospholipids associated with birth weight. Open circles represent all phospholipids (PL) and lysophospholipid (LPL) species. Colored circles represent PL and LPL containing odd and branched chain fatty acids. Red circles represent non-significant associations, blue circles represent significant lipids (p < 0.05), green circles represent 10 most significant lipids (lowest p-values)

Figure legends can be improved for clarity. For example, for p-values, 10-15 should be used, rather than E-15.

Figure legends have now been modified to explain the forest plots. P-values have been changed as recommended

Figure 2 The newborn cord serum lipidome is influenced by gestational age (GA) and birth weight (BW). a, Estimated percentage difference of the lipid species per day increase in gestational age, determined using linear regression, adjusted for weight, sex, mode of birth, duration of labor, maternal pre-pregnancy BMI, GDM, maternal education, birth order and lipidomics run batch. b, Estimated percentage difference of the individual lipid species per kilogram increase in birthweight, determined using the same model **as a.** Each circle on the plot represents a lipid species, grey open circles show non-significant lipid species (p>0.05), white closed circles show significant lipid species (p < 0.05), the top 10 lipid species (10 lowest p values) are shown in blue circles. Each red diamond represents a lipid class. All p-values were corrected for multiple comparisons. The horizontal bars (error bars) are shown for the significant lipid species (white closed circles), the top ten lipid species (blue circles), corrected p-value < 2.88 x 10^-24^ and 1.81 x 10^-15^ respectively and the lipid classes (red diamonds). The Error bars represent 95% confidence intervals. Grey dotted lines separate sphingolipids, phospholipids and other lipids.

Red diamonds are said to represent lipid classes – but what does this mean exactly? Is it the mean or median value of all species within the class?

Red diamonds represent lipid class totals, calculated as the sum of lipid species in that particular lipid class. This has been explained in the methods section (page 7, line 16 – 20).

Lipid class total concentrations were calculated as the sum of individual lipid species concentrations within the class, except in the case of triacylglycerol (TGs) and alkyl-diacylglycerol (TG-Os), where we measured both neutral loss [NL] and single ion monitoring [SIM] peaks, and subsequently used the more accurate, but less structurally resolved, [SIM] species concentrations for summation purposes when examining lipid class totals

In Supplementary Figure 5, the legend title says, "Association of infant BMI with lipid species". However, panel d relates to 4-year-old children i.e., toddlers, rather than infants; the correct word to use is "child".

This figure has been deleted

Other Suggestions and Points Requiring Clarification:The authors note the interesting observation that acylcarnitine, DAGs and TAGs were more abundant with longer duration of labor, alluding to a role for fueling the strenuous activity of labor. How does one know that the increased levels of these lipid species are due to the physiology of labor per se (on page 22, the authors write, "stress in utero and during birth increases energy mobilization for high energy expenditure in these modes of birth"), rather than a result of mothers often being prohibited from consuming food and drink (essentially an enforced state of fasting) during labor?

The reviewer has made a valid point and the fasted state of the mother may also contribute to the response of the infant, potentially as a result of decreased circulating fatty acids available for transfer across the placenta.

We have included this possibility in the discussion (page 31 line 16-19)

“However, a decrease in circulating free fatty acids and triacylglycerols in scheduled CS may also be a result of mothers often being prohibited from consuming food and drink before the procedure leading to a fasting state and so decreased availability of free fatty acids for transfer across the placenta.”

A few clinical issues related to breastfeeding and child lipidomic changes could have been but were also not addressed by the authors using the present dataset. For example, it would be important to know whether the mode of feeding – exclusive breastfeeding or mixed feeding – influences the development of the child's lipidome. Other questions that can be answered as well include: (a) delineating the influences of different types of formula milk (cow versus goat, for example) on the child's lipidome at different time points, (b) definition of particular breast milk lipid signatures that correlate to higher infant/toddler BMI, and (c) determination of breast milk lipid content and compositions for GDM and healthy mothers.

The authors agree that breast feeding is an important factor in the infant lipidome. We have now included an analysis of the association of breast feeding with plasma lipids at 6 and 12 months of age and observed strong associations with most lipid species and classes.

This has been included in the results page 26 line 1-15, Figure 6a and 6b, Supplementary Tables 13 and 14. We further discuss how breastfeeding can influence the lipid metabolism in the first year of life page 34, line 3 to page 35, line 7.

Please refer to Essential revisions 4

We also agree that the analyses proposed above are important and we are currently analysing breast milk samples and performing additional analyses with these data to address these issues. However this is outside the scope of this manuscript.